# The worldwide burden of HIV in transgender individuals: An updated systematic review and meta-analysis

Sarah E. Stutterheim*, Mart van Dijk, Haoyi Wang, Kai J. Jonas

Department of Work and Social Psychology, Maastricht University, Maastricht, The Netherlands

* s.stutterheim@maastrichtuniversity.nl

## Abstract

### Introduction

Transgender individuals are at risk for HIV. HIV risks are dynamic and there have been substantial changes in HIV prevention (e.g., pre-exposure prophylaxis [PrEP]). It is thus time to revisit HIV prevalence and burden among transgender individuals. The objective of this systematic review and meta-analysis was thus to examine worldwide prevalence and burden of HIV over the course of the epidemic among trans feminine and trans masculine individuals.

### Methods

We conducted an updated systematic review by searching PsycINFO, PubMed, Web of Science, and Google Scholar, for studies of any research design published in in a peer-reviewed journal in any language that reported HIV prevalence among transgender individuals published between January 2000 and January 2019. Two independent reviewers extracted the data and assessed methodological quality. We then conducted a meta-analysis, using random-effects modelling, to ascertain standardized prevalence and the relative burden of HIV carried by transgender individuals by country and year of data collection, and then by geographic region. We additionally explored the impact of sampling methods and pre-exposure prophylaxis (PrEP).

### Results

Based on 98 studies, overall standardized HIV prevalence over the course of the epidemic, based on weights from each country by year, was 19.9% (95% CI 14.7% - 25.1%) for trans feminine individuals (n = 48,604) and 2.56% (95% CI 0.0% - 5.9%) for trans masculine individuals (n = 6460). Overall OR for HIV infection, compared with individuals over age 15, was 66.0 (95% CI 51.4–84.8) for trans feminine individuals and 6.8 (95% CI 3.6–13.1) for trans masculine individuals. Prevalence varied by geographic region (13.5% - 29.9%) and sampling method (5.4% - 37.8%). Lastly, PrEP effects on prevalence could not be established.

**Funding:** This systematic review and meta-analysis was funded by AIDSFonds (P-30805). See https://aidsfonds.nl. SS & KJ received the funding. The funders had no role in study design, data collection and analysis, decision to publish, or preparation of the manuscript.

**Competing interests:** The authors have declared that no competing interests exist.

## Conclusion

Trans feminine and trans masculine individuals are disproportionately burdened by HIV. Their unique prevention and care needs should be comprehensively addressed. Future research should further investigate the impact of sampling methods on HIV prevalence, and monitor the potential impact of PrEP.

## Introduction

Transgender individuals, defined as individuals who experience a misalignment between the sex they were assigned at birth and their gender identity or whose gender identity is incongruent with gender norms, [1, 2] are at significant risk for an HIV infection [3]. Individual level risk factors include condomless sex, particularly receptive anal sex, coinfection with other sexually transmitted infections, transactional sex, and the shared use of needles for hormone and/or silicon injections [4–8]. Individual level risk factors do not stand alone; they result from, and intersect with, other factors such as mental health difficulties, substance use, and many forms of marginalization and stigmatization that limit, among other things, educational and work opportunities, as well as legal recognition of one's chosen gender [6, 9–14]. Given that HIV risk among transgender individuals is a dynamic phenomenon, it is important to regularly monitor and update our knowledge of HIV prevalence and burden, such that we can identify trends that can inform policy-making and interventions. Here, we present a comprehensive updated systematic review of HIV prevalence over the course of the epidemic and a meta-analyses of HIV burden among transgender individuals covering literature from 2000 until 2019.

### Previous systematic reviews and meta-analyses

Since 2008, a series of systematic reviews and meta-analyses have been published [1, 4, 5, 11, 15–17]. The first, by Herbst and colleagues, [15] investigated HIV prevalence among trans individuals in the United States, covering literature from 1988 until early 2007, and included laboratory-confirmed and self-reported prevalence. Pooled HIV prevalence based upon studies reporting laboratory-confirmed HIV status was, for trans women, 27.7%. HIV prevalence among trans women based upon self-reported HIV status was 11.8%. Among trans men, only one study reported laboratory confirmed prevalence (2%) and self-reported prevalence rates ranged from 0% to 3%. The second systematic review and meta-analysis by Operario et al. [16] set out to assess whether transgender female sex workers (FSW) experienced higher HIV infection rates than cis-gender sex workers and transgender women who do not engage in sex work, using both laboratory-confirmed and self-reported HIV prevalence rates published between 1998 and 2006. HIV prevalence was 27.3% in transgender FSW and 14.7% in trans women who did not engage in sex work. Operario et al.'s meta-analysis further showed that transgender FSW are at significantly higher risk for HIV than cis-gender sex workers and trans women who do not engage in sex work [16]. In 2013, Baral and colleagues [4] published a systematic review and meta-analysis of HIV prevalence among transgender women, covering literature from 2000 to November 2011 and using only laboratory-confirmed HIV prevalence rates. Pooled prevalence was 19.1% and the meta-analytical findings showed that, compared to all adults of reproductive age, the odds ratio for HIV infection in trans women was 49 across the 15 countries included, thus demonstrating that transgender women carry a high burden of HIV [4]. Poteat and colleagues [5] followed up on Baral et al.'s work with a systematic review, but not a meta-analysis, of HIV

prevalence literature published between 2012 and 2015, looking now at both trans feminine and trans masculine populations. Prevalence rates varied substantially based on locale but Poteat et al. [5] concluded, in line with the previous reviews, that HIV prevalence was high in trans feminine populations. They also concluded that data on HIV among trans masculine individuals is still very limited. Almost simultaneously, Reisner and Murchison [1] published a global research synthesis of HIV and STI risks in adult trans men, but not trans women, using 25 studies. They found HIV prevalence rates for trans men ranging from 0% to 4.3% for laboratory-confirmed HIV status and 0% to 10% for self-reported HIV, suggesting that trans masculine individuals may also be more vulnerable to HIV than cis-gender adults. Additionally, in 2017, MacCarthy and colleagues [11] published a global systematic review of HIV and sexually transmitted infections among transgender individuals. They reported HIV prevalence rates ranging from 0% to 17.6% for self-reported HIV status and 0.6% to 34.1% for laboratory-confirmed HIV status. However, given their focus on HIV and STI co-infection, the HIV prevalence rates reported in that review were derived only from studies that also reported STI prevalence rates. The reported prevalence rates were thus based on a mere 6 studies for self-reported HIV status and 13 studies for laboratory-confirmed HIV status. Recently, Becasen et al. [17] published a systematic review and meta-analysis of HIV prevalence among transgender individuals in the United States only using literature published in the United States between 2006 and May 2017. They established that laboratory-confirmed HIV prevalence was 14.1% for trans women and 3.2% for trans men; self-reported prevalence was 16.1% and 1.2% for trans women and trans men, respectively.

## Current concerns

Overall, the various systematic reviews and meta-analyses demonstrate that transgender individuals, particularly trans feminine individuals, are disproportionately burdened with HIV but none of the more recent systematic reviews have comprehensively updated Baral et al.'s worldwide systematic review and meta-analysis with both transfeminine and transmasculine individuals. Furthermore, from a methodological perspective, more fine-grained analyses (e.g. by country and year of data collection) are being called for, rather than only pooled analyses by country or region, as has been the methodological approach in previous meta-analyses. Additionally, critique about reported prevalence rates has been levied, with the claim that many studies have relied on convenience samples of, often, transgender women who engage in sex work, which may inflate prevalence rates [18, 19]. Also, previous meta-analyses have not differentiated between various sampling strategies and this may impact meta-analytical findings. Further, there have been substantial and fundamental changes in HIV prevention in recent years. One is the emergence of pre-exposure prophylaxis (PrEP) as a powerful tool for HIV prevention for at risk groups like transgender individuals [5, 8, 20–23]. With these considerations in mind, we feel it is time to revisit worldwide HIV prevalence and burden among transgender individuals. We therefore systematically reviewed literature published between 2000 and 2019 on HIV prevalence among transgender individuals and then conducted a meta-analysis 1) to establish prevalence rates for both trans feminine and trans masculine individuals; and 2) to compare the burden of HIV infection among transgender individuals to individuals over 15 years of age in the countries and regions from which samples were derived, taking year of data collection into account. We then explored the possible impact of sampling methods and of PrEP on prevalence rates and the burden of HIV infection.

## Methods

### Search strategies and eligibility

We searched, in November 2017 and again in January 2019, PsycINFO, PubMed, Web of Science, and Google Scholar®, for studies in all languages published between January 1st, 2000

and January 28[th], 2019. We selected this timeframe in order to gain a complete, comprehensive, and nuanced understanding of worldwide prevalence over and burden of HIV among transgender individuals. We also reviewed the studies included in Baral et al. [4] and in Poteat et al. [5] to ensure that they were covered in our analysis as well. We explicitly overlapped the timeframe in our meta-analysis with those of previous meta-analyses in order to generate comprehensive and robust meta-analytical findings. It also allowed us to explore the impact of applying more refined methodology (standardized vs. pooled prevalence rates) in the meta-analysis, and compare findings delivered by the different meta-analytical approaches. Articles and citations were downloaded and managed in the reference software Mendeley®.

We searched for articles on (the treatment of) HIV and transgender individuals using the following search terms: HIV OR AIDS OR "PrEP" OR "Pre-Exposure Prophylaxis" OR "TasP" OR "treatment as prevention" AND *transgender* OR "MTF" or "male to female transgender" OR "FTM" OR "female to male transgender" OR *transsexual* OR "travesty" OR "cross dresser" OR "koti" OR "hijra" OR "mahuvahine" OR "mahu" OR "waria" OR "katoey" OR "bantut" OR "nadleehi" OR "berdache" OR "xanith". These terms are in line with the terms previously used by Baral et al. [4].

Studies of any research design published in peer-reviewed journals that reported laboratory- confirmed prevalence of HIV among transgender individuals were included. When prevalence rates were pooled across trans feminine and trans masculine individuals or when prevalence was pooled across trans feminine individuals and men who have sex with men (MSM), we contacted the authors and requested separate prevalence rates for the populations included.

## Study selection and data extraction

Titles and abstracts were screened by two independent reviewers and articles that clearly did not include HIV prevalence data were excluded, as were duplicates. All articles that met the inclusion criteria and articles that needed further review to ascertain whether they met the criteria were subsequently downloaded. When one reviewer deemed the title and/or abstract potentially relevant and the other did not, the full-text for that article was nonetheless downloaded. Subsequently, the full texts were reviewed. When studies reported duplicate data, the study with the smallest sample size was excluded. If sample sizes were identical, the later publication was excluded. Any conflicts over study inclusion were resolved by project leads (KJ and SS) in conjunction with the researchers running the meta-analysis (MvD and HW). The PRISMA reporting checklist was used to guide the reporting of this study. No protocol was registered for this review.

Data were extracted by two trained coders using a standardized extraction form that included details about sample size, sampling method, sample description, recruitment location, time period of study, age range, transgender type (trans masculine/trans feminine/both), HIV measure (self-reported/laboratory testing), and HIV prevalence or incidence.

## Methodological quality assessment

Given the lack of consensus on fitting quality assessment tools for epidemiological studies, [24] we developed criteria specifically for this systematic review and meta-analysis. In doing so, we used and adapted appropriate criteria from the JBI Critical Appraisal Checklist for Studies Reporting Prevalence Data [25]. Studies were deemed of sufficient quality if: 1) biological testing (rather than self-reported HIV status) was used to establish HIV diagnoses, as was done in Baral et al. [4]; 2) study participants were described in sufficient detail, meaning that prevalence was reported, or subsequently obtained directly from the authors, specifically for

trans feminine and/or trans masculine individuals (rather than transgender individuals as a whole group); 3) if the study setting/location was sufficiently detailed; 4) if the data collection timeframe was reported; 5) if prevalence or frequency of HIV diagnosis within the total sample were reported; and 6) if sample size was at least 40 for trans feminine individuals. We did not apply a minimum sample size for studies reporting prevalence among trans masculine individuals as the majority of studies had small sample sizes, and a minimum sample size would have led to the exclusion of most studies reporting HIV prevalence among trans masculine individuals. Additionally, we did not exclude studies based on sampling method as investigating the impact of sampling methods was one of the objectives of this meta-analysis.

## Data analyses

First, we used analogous methodology to prior meta-analyses [4, 26, 27]. We grouped studies by country, weighted by sample size. We calculated pooled HIV prevalence and 95% confidence intervals (CIs) per country. We did this separately for trans feminine and trans masculine samples. In line with previous meta-analyses, we then calculated odds ratios per country by dividing the HIV prevalence among transgender individuals (numerator) by the HIV prevalence rate among individuals over 15 years of age in the general population in the country from which the sample was derived (denominator), as reported by the 2017 UNAIDS reports (where prevalence estimates for adults are from 15 years of age onward) [28] and estimations of adult population size from the US Census Bureau International Division [29]. These results are reported in S1 Appendix.

Then, to achieve a more refined methodological analysis, we standardized rather than pooled prevalence rates, and ran the meta-analysis again, this time matching country-level prevalence rates to year(s) of data collection for the included studies. When data were collected over multiple years in the original studies, the median year of the year-span was chosen for the country by year analysis. If HIV prevalence in the sample was 0, we calculated confidence intervals using the Wilson interval [30]. Then, we grouped countries by geographic region (Africa, Latin America, Asia, and Global North) and calculated, per geographic region, the standardized HIV prevalence among trans feminine individuals as well as odds ratios based on weights from each country-year.

Subsequently, given recent discussions about the impact of sampling methods on findings pertaining to HIV prevalence among trans feminine individuals, [18] we grouped studies by sampling method, and calculated standardized HIV prevalence by sampling method. We delineated ten sampling methods, namely cluster sampling, convenience sampling, purposive sampling, respondent driven sampling, snowball sampling, sampling from database health plan, as well as sampling via STI clinic, via hospital, via NGO, and via surveillance. Overlap in categories may exist as some studies used multiple sampling methods. In such cases, we categorized the study under its primary sampling method.

Lastly, we explored possible effects of the introduction of PrEP on HIV prevalence among trans feminine individuals. We focused on US studies only as PrEP has been available in the US since 2012, which is longer than in any other country. We conducted subgroup analyses, with data being collected either prior to the introduction of PrEP (1997–2011), or after the introduction of PrEP (2012–2017).

The meta-analysis was conducted with the statistical software R [31] using the metafor package [32]. We used a random-effects model and the DerSimonian-Laird method to estimate the model. The DerSimonian-Laird $Q$ test and $I^2$ values were used to assess heterogeneity, with low, moderate, and high heterogeneity corresponding to $I^2$ values of 25%, 50%, and 75%. [33] We investigated publication bias by inspecting funnel plots [34].

## Results

The study selection process is presented in Fig 1. We included 98 studies from a total of 34 countries, of which 78 studies described HIV prevalence in trans feminine individuals, 4 described prevalence in trans masculine individuals, and 16 described both. In total, we included 48,604 trans feminine individuals from 34 countries and 6460 trans masculine individuals from 5 countries. The included studies and relevant characteristics of those studies are reflected in Table 1.

The overall standardized HIV prevalence over the course of the epidemic, based on weights from each country by year, was 19.9% (95% CI 14.7% - 25.1% Table 2) for trans feminine individuals and 2.56% (95% CI 0.0% - 5.9%; Table 3) for trans masculine individuals. The overall OR for HIV infection, compared with individuals over 15 years of age, was 66.0 (95% CI 51.4–84.8; Table 2 and Fig 2) for trans feminine individuals and 6.8 (95% CI 3.6–13.1, Table 3 and Fig 3) for trans masculine individuals. Tables 2 and 3 also show the overall standardized prevalence rates and overall odds ratios per country by year for trans feminine individuals and trans masculine individuals, respectively.

Standardized prevalence rates and overall odd ratios (based on weights from each country by year) according to geographic region are presented in Table 4. In sub-Saharan Africa (*n* = 1192), standardized HIV prevalence among trans feminine individuals was 29.9% (95% CI 22.5% - 37.3%) and the overall OR for HIV infection, compared to individuals over 15

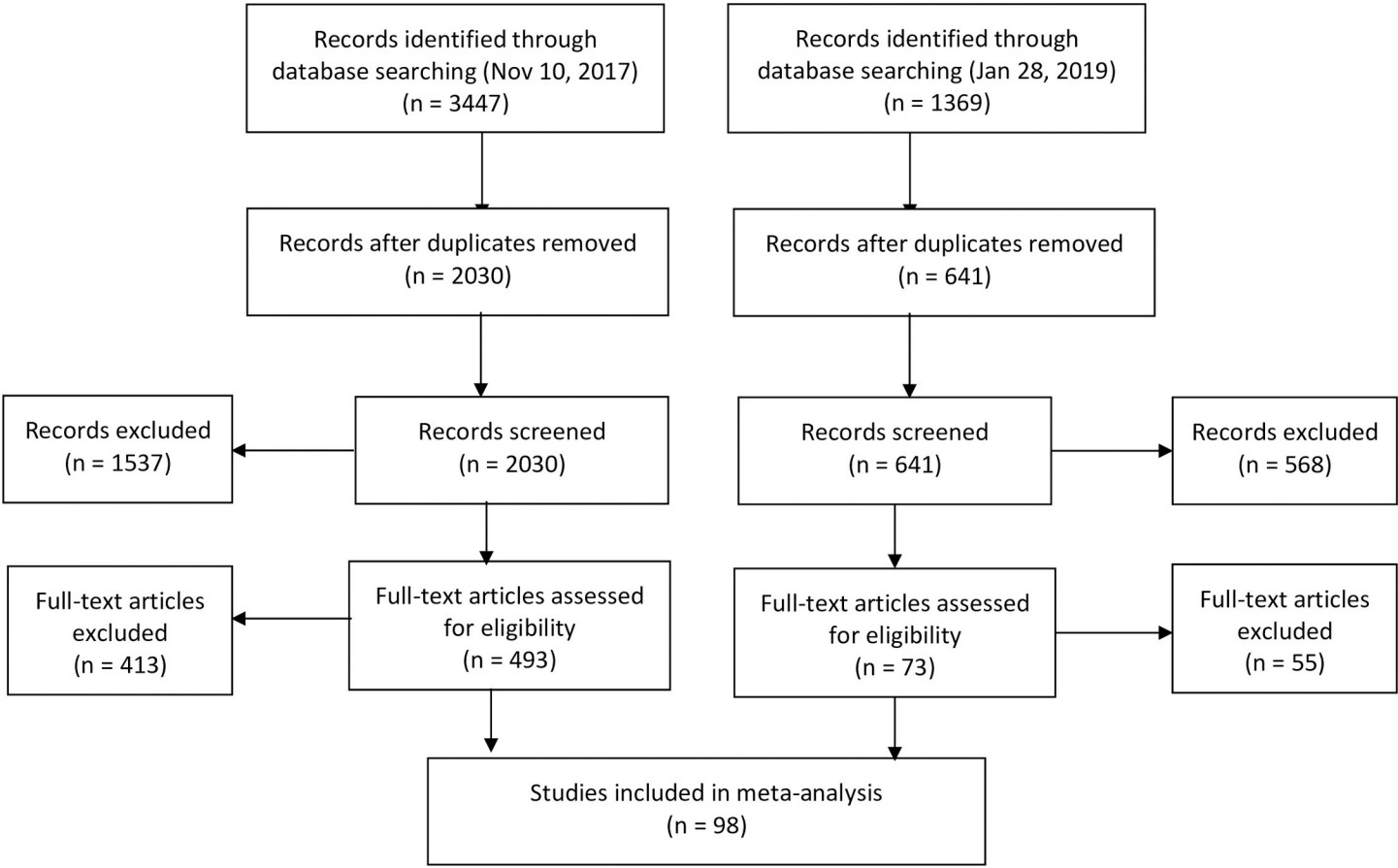

**Fig 1. PRISMA flow chart describing the study selection process.**

**Table 1. Studies included in review and meta-analysis.**

| Authors | Year of publication | Year of data collection | Transgender sample | HIV prevalence (%) | HIV frequency (n) | Sample size | Country | Geographic region | Sampling method |
|---|---|---|---|---|---|---|---|---|---|
| Aguayo, Munoz, & Aguilar [35] | 2013 | 2011 | TF | 27.00% | 64 | 237 | Paraguay | Latin America | Cluster sampling |
| Akhtar, Badshah, Akhtar, et al. [36] | 2012 | 2009–2010 | TF (hijras) | 21.60% | 66 | 306 | Pakistan | Asia | Respondent driven sampling |
| Altaf [37] | 2009 | 2006–2007 | TF (hijras) | 4.70% | 38 | 810 | Pakistan | Asia | Surveillance |
| Altaf, Zahidie, & Agha [38] | 2012 | 2008 | TF (hijras) | 6.40% | 75 | 1181 | Pakistan | Asia | Surveillance |
| Baqi, Shah, Baig et al. [39] | 2006 | 1998 | TF (hijras) | 0.00% | 0 | 208 | Pakistan | Asia | Respondent driven sampling |
| Barrington, Weijnert, & Guardado et al. [40] | 2012 | 2008 | TF | 19.00% | 13 | 67 | El Salvador | Latin America | Respondent driven sampling |
| Bastos, Bastos, Coutinho et al. [41] | 2018 | 2016–2017 | TF | 29.62% | 843 | 2846 | Brazil | Latin America | Respondent driven sampling |
| Bellhouse, Walker, Fairley et al. [42] | 2016 | 2011–2014 | TM | 3.57% | 1 | 28 | Australia | Global North | STI clinic visit |
| | | | TF | 10.39% | 8 | 77 | | | |
| Brahmam, Kodavallaa, Rajkumar et al. [43] | 2008 | 2006–2007 | TF (hijras) | 18.10% | 104 | 575 | India | Asia | Cluster sampling |
| Carballo-Dieguez, Balan, Dolezal et al. [44] | 2012 | 2005–2006 | TF | 13.00% | 12 | 84 | Brazil | Latin America | Respondent driven sampling |
| Castel, Magnus, Peterson et al. [45] | 2012 | 2006 | TF & TM | 10.59% | 9 | 85 | US | Global North | STI clinic visit |
| Castillo, Konda, Leon et al. [46] | 2015 | 2008–2009 | TF | 16.82% | 35 | 208 | Peru | Latin America | Snowball |
| Chariyalersak, Kosachunhanan, Saokhieo et al. [47] | 2011 | 2008–2009 | TF | 9.30% | 13 | 140 | Thailand | Asia | STI clinic visit |
| Chen, McFarland, Tompson et al. [48] | 2011 | 2009 | TM | 0.00% | 0 | 59 | US | Global North | STI clinic visit |
| Chhim, Ngin, Chhoun et al. [49] | 2017 | 2015–2016 | TF | 5.90% | 81 | 1375 | Cambodia | Asia | Respondent driven sampling |
| Clements-Noelle, Wilkenson, Kitano et al. [50] | 2001 | 1997 | TM | 2.00% | 2 | 123 | US | Global North | Respondent driven sampling |
| | | | TF | 35.00% | 137 | 392 | | | |
| Colby, Nguyen, Le et al. [51] | 2016 | 2015 | TF | 18.00% | 37 | 205 | Vietnam | Asia | Snowball |
| Costa, Fontanari, Jacinto et al. [52] | 2015 | 1998–2014 | TM | 25.00% | 0 | 51 | Brazil | Latin America | Hospital |
| | | | TF | 25.00% | 71 | 284 | | | |
| Dasarathan & Kalaivani [53] | 2017 | 2011–2014 | TF | 13.40% | 11 | 82 | India | Asia | STI clinic visit |
| Diez, Bleda, Varela et al. [54] | 2014 | 2000–2009 | TF | 24.50% | 129 | 529 | Spain | Global North | STI clinic visit |
| Dos Ramos Farias, Garcia, Reynaga et al. [55] | 2011 | 2006–2009 | TF | 34.10% | 93 | 273 | Argentina | Latin America | Respondent driven sampling |
| Fernandes, Zanini, Rezende et al. [56] | 2015 | 2011–2013 | TF | 24.34% | 37 | 152 | Brazil | Latin America | Cluster sampling |
| Fernandez-Balbuena, Belza, Urdaneta et al. [57] | 2015 | 2008–2012 | TF & TM | 45.54% | 46 | 101 | Spain | Global North | NGO |
| Fernandez-Lopez, Reyes-Uruena, Agusti et al. [58] | 2018 | 2014–2016 | TF | 8.83% | 40 | 453 | Spain | Global North | STI clinic visit |

*(Continued)*

**Table 1.** (Continued)

| Authors | Year of publication | Year of data collection | Transgender sample | HIV prevalence (%) | HIV frequency (n) | Sample size | Country | Geographic region | Sampling method |
|---|---|---|---|---|---|---|---|---|---|
| Grandi, Goihman, Ueda et al. [59] | 2000 | 1992–1998 | TF | 40.00% | 174 | 434 | Brazil | Latin America | Respondent driven sampling |
| Green, Hoenigl, Morris et al. [60] | 2015 | 2008–2014 | TM | 3.00% | 1 | 30 | US | Global North | STI clinic visit |
| | | | TF | 2.00% | 3 | 151 | | | |
| Grinsztejn, Jalil, Monteiro et al. [61] | 2017 | 2015–2016 | TF | 31.20%/ 24.20% | 141 | 345 | Brazil | Latin America | Respondent driven sampling |
| Guadamuz, Wimonsate, Varangrat et al. [62] | 2011 | 2005 | TF | 14.00% | 64 | 474 | Thailand | Asia | Convenience sampling |
| Gutierrez, Tajada, Alvarez et al. [63] | 2004 | 1998–2003 | TF | 23.00% | 14 | 60 | Spain | Global North | Convenience sampling |
| Guy, Mustikawati, Wijaksono et al. [64] | 2011 | 2006–2008 | TF & TM | 31.60% | 151 | 477 | Indonesia | Asia | STI clinic visit |
| Habarta, Wang, Mulatu et al. [65] | 2015 | 2009–2011 | TM | 0.51% | 12 | 2364 | US | Global North | STI clinic visit |
| | | | TF | 2.70% | 355 | 13154 | | | |
| Hadikusumo, Utsumi, Amin et al. [66] | 2016 | 2012 | TF | 16.00% | 16 | 100 | Indonesia | Asia | STI clinic visit |
| Hakim, Coy, Patnaik et al. [67] | 2018 | 2014–2015 | TF | 22.42% | 37 | 165 | Mali | Africa | Respondent driven sampling |
| Hawkes, Collumbien, Platt et al. [68] | 2009 | 2007 | TF (khusra) | 2.00% | 6 | 269 | Pakistan | Asia | Respondent driven sampling |
| Hiransuthikul, Pattanachaiwit, Teeratakulpisarn et al. [69] | 2018 | 2012–2013 | TF | 4.26% | 2 | 47 | Thailand | Asia | STI clinic visit |
| Januraga, Wulandari, Muliawan et al. [70] | 2013 | 2009–2010 | TF (waria) | 36.87% | 80 | 217 | Indonesia | Asia | Respondent driven sampling |
| Jin, Restar, Biello et al. [71] | 2019 | 2012–2015 | TF | 24.71% | 65 | 263 | US | Global North | Convenience sampling |
| Kaplan, McGowan, & Wagner [72] | 2016 | 2012 | TF | 10.00% | 4 | 40 | Lebanon | Asia | Respondent driven sampling |
| Kellogg, Clements-Nolle, Dilley et al. [73] | 2001 | 1997–2000 | TF | 15.00% | 37 | 238 | US | Global North | STI clinic visit |
| Keshinro, Crowell, Nowak et al. [74] | 2016 | 2013–2016 | TF | 71.43% | 75 | 105 | Nigeria | Africa | Respondent driven sampling |
| Khan, Rehan, Qayyum et al. [75] | 2008 | 2004 | TF (hijras) | 1.00% | 5 | 409 | Pakistan | Asia | Cluster sampling |
| Kojima, Park, Konda et al. [76] | 2017 | 2013–2014 | TF | 30.10% / 27.60% | 30 | 89 | Peru | Latin America | STI clinic visit |
| Leinung, Urizar, Patel et al. [77] | 2013 | prior 2003 | TM | 0.00% | 0 | 50 | US | Global North | Hospital |
| | | | TF | 8.33% | 16 | 192 | | | |
| Lipsitz, Segura, Castro et al. [78] | 2014 | 2007–2009 | TF | 30.00% | 64 | 214 | Peru | Latin America | STI clinic visit |
| Lobato, Koff, Schestatsky et al. [79] | 2008 | 1998–2005 | TM | 0.00% | 0 | 16 | Brazil | Latin America | Hospital |
| | | | TF | 19.67% | 24 | 122 | | | |

*(Continued)*

**Table 1.** (Continued)

| Authors | Year of publication | Year of data collection | Transgender sample | HIV prevalence (%) | HIV frequency (n) | Sample size | Country | Geographic region | Sampling method |
|---|---|---|---|---|---|---|---|---|---|
| Logie, Lacombe-Duncan, Wang et al. [80] | 2016 | 2015 | TF | 25.20% | 26 | 103 | Jamaica | Latin America | Respondent driven sampling |
| Long, Montano, Cabello et al. [81] | 2017 | 2013–2015 | TF | 19.68% | 61 | 310 | Peru | Latin America | STI clinic visit |
| Luzzati, Zatta, Pavan et al. [82] | 2016 | 2000–2014 | TM | 0.00% | 0 | 20 | Italy | Global North | Hospital |
| | | | TF | 12.10% | 21 | 173 | | | |
| Manieri, Castellano, Crespi et al. [83] | 2014 | 2005–2011 | TM | 0.00% | 0 | 27 | Italy | Global North | Hospital |
| | | | TF | 5.36% | 3 | 56 | | | |
| McFarland, Wilson, Raymond et al. [84] | 2017 | 2014 | TM | 0.00% | 0 | 122 | US | Global North | Convenience sampling |
| Mimiaga, Hughto, Biello et al. [85] | 2019 | 2012–2015 | TF | 20.60% | 48 | 233 | US | Global North | Convenience sampling |
| Murrill, Liu, Guilin et al. [86] | 2008 | 2004 | TF & TM | 13.00% | 9 | 92 | US | Global North | Convenience sampling |
| Nemoto, Bödeker, Iwamoto et al. [87] | 2014 | 2000–2001 | TF | 29.93% | 161 | 538 | US | Global North | Purposive sampling |
| Nguyen, Nguyen, Le et al. [88] | 2008 | 2004 | TF ("male transvestites" "bong lo") | 7.00% | 5 | 75 | Vietnam | Asia | Convenience sampling |
| Nuttbrock, Bockting, Rosenblum et al. [89] | 2013 | 2004–2007 | TF | 2.80% | 9 | 230 | US | Global North | Convenience sampling |
| Nuttbrock, Hwahng, Bockting et al. [90] | 2009 | earlier than 2009 | TF | 35.98% | 186 | 517 | US | Global North | Convenience sampling |
| Ongwandee, Lertpiriyasuwat, Khawcharoenporn et al. [91] | 2018 | 2015–2016 | TF | 900.00% | 39 | 435 | Thailand | Asia | STI clinic visit |
| Pando, Gomez-Carrillo, Vignoles et al. [92] | 2011 | 2006–2008 | TF | 34.00% | 38 | 112 | Argentina | Latin America | NGO |
| Patrascioiu, Lopez, Porta et al. [93] | 2013 | 2006–2010 | TM | 2.20% | 2 | 92 | Spain | Global North | Convenience sampling |
| | | | TF | 12.60% | 18 | 142 | | | |
| Peitzmeier, Reisner, Harigopal et al. [94] | 2014 | 2006–2012 | TM | 0.86% | 2 | 233 | US | Global North | Hospital |
| Pell, Prone, Vlahakis et al. [95] | 2011 | 2004 | TM | 0.00% | 0 | 17 | Australia | Global North | STI clinic visit |
| | | | TF | 4.26% | 6 | 141 | | | |
| Pisani, Girault, Gultom et al. [96] | 2004 | 2002 | TF (waria) | 22.00% | 53 | 241 | Indonesia | Asia | Cluster sampling |
| Pitasi, Oraka, Clark et al. [97] | 2019 | 2010–2013 | TM | 8.30% | 10 | 120 | US | Global North | STI clinic visit |
| | | | TF | 14.20% | 72 | 506 | | | |
| Pizzicato, Vagenas, Gonzales et al. [98] | 2017 | 2011 | TF | 14.59% | 104 | 713 | Peru | Latin America | Respondent driven sampling |
| Poteat, Ackerman, Diouf et al. [99] | 2017 | 2011–2016 | TF | 2.78% | 3 | 108 | Burkina Faso | Africa | Respondent driven sampling |
| | | | TF | 25.50% | 76 | 298 | Côte d'Ivoire | | |
| | | | TF | 59.15% | 42 | 71 | Lesotho | | |
| | | | TF | 16.00% | 12 | 75 | Malawi | | |
| | | | TF | 37.19% | 74 | 199 | Senegal | | |
| | | | TF | 14.17% | 17 | 120 | Swaziland | | |
| | | | TF | 17.65% | 9 | 51 | Togo | | |

(*Continued*)

**Table 1.** (*Continued*)

| Authors | Year of publication | Year of data collection | Transgender sample | HIV prevalence (%) | HIV frequency (n) | Sample size | Country | Geographic region | Sampling method |
|---|---|---|---|---|---|---|---|---|---|
| Poteat, German, & Flynn [100] | 2016 | 2004–2005 | TF | 43.00% | 21 | 49 | US | Global North | Surveillance |
| Prabawanti, Bollen, Palupy et al. [101] | 2011 | 2007 | TF (waria) | 24.40% | 183 | 748 | Indonesia | Asia | Cluster sampling |
| Quinn, Nash, Hunkeler et al. [102] | 2017 | 2006–2014 | TM | 0.31% | 9 | 2892 | US | Global North | Database health plan |
| | | | TF | 5.35% | 186 | 3475 | | | |
| Rana, Reza, Alam et al. [103] | 2016 | 2012 | TF (hijras) | 0.80% | 7 | 889 | Bangladesh | Asia | STI clinic visit |
| Raymond, Wilson, Packer et al. [104] | 2019 | 2010 | TF | 39.17% | 123 | 314 | US | Global North | Respondent driven sampling |
| | | 2013 | TF | 36.05% | 84 | 233 | | | |
| | | 2016 | TF | 38.68% | 123 | 318 | | | |
| Reback, Lombardi, Simon et al. [105] | 2005 | 1998–1999 | TF | 22.10% | 54 | 244 | US | Global North | STI clinic visit |
| Reisner, White, Mayer et al. [106] | 2014 | 2007 | TM | 4.35% | 1 | 23 | US | Global North | STI clinic visit |
| Reisner, Vetters, White et al. [107] | 2015 | 2001–2010 | TF | 7.93% | 5 | 63 | US | Global North | STI clinic visit |
| | | | TM | 2.40% | 2 | 82 | | | |
| Rich, Scott, Johnston, et al. [108] | 2017 | 2012–2014 | TM | 0.00% | 0 | 11 | Canada | Global North | Respondent driven sampling |
| Rowe, Santos, McFarland et al. [109] | 2015 | 2012–2013 | TF | 4.00% | 13 | 292 | US | Global North | Snowball |
| Russi, Serra, Vinoles et al. [110] | 2003 | 1999 | TF ("male transvestites") | 21.50% | 49 | 200 | Uruguay | Latin America | Convenience sampling |
| Sahastrabuddhe, Gupta, Stuart et al. [111] | 2012 | 1993–2002 | TF (hijras) | 45.20% | 38 | 84 | India | Asia | STI clinic visit |
| Salas-Espinoza, Menchaca-Diaz, Patterson et al. [112] | 2017 | 2012 | TF | 22.00% | 22 | 100 | Mexico | Latin America | Cluster sampling |
| Saravanamurthy, Rajendran, Ramakrishnan et al. [113] | 2008 | 2007 | TF | 17.50% | 23 | 125 | India | Asia | Respondent driven sampling |
| Schulden, Song, Barros et al. [114] | 2008 | 2005–2006 | TM | 0.00% | 0 | 42 | US | Global North | Convenience sampling |
| | | | TF | 12.00% | 67 | 559 | | | |
| Seekaew, Pengnonyang, Jantarapakde et al. [115] | 2018 | 2015–2016 | TF | 8.80% | 69 | 786 | Thailand | Asia | Respondent driven sampling |
| Shan, Yu, Yang et al. [116] | 2018 | 2016 | TF | 7.60% | 38 | 498 | China | Asia | Snowball |
| Shaw, Lorway, Bhattacharjee et al. [117] | 2016 | 2011 | TF (kothi & hijras) | 15.30% | 27 | 176 | India | Asia | Cluster sampling |
| Shaw, Emmanuel, Adrien et al. [118] | 2011 | 2005–2006 | TF (hijras) | 1.00% | 10 | 1162 | Pakistan | Asia | Cluster sampling |
| Sherman, Park, Galai et al. [119] | 2019 | 2016–2017 | TF | 40.30% | 25 | 62 | US | Global North | Convenience sampling |
| Shinde, Setia, Row-Kavi et al. [120] | 2009 | earlier than 2009 | TF | 41.00% | 21 | 51 | India | Asia | STI clinic visit |
| Silva-Santisteban, Raymond, Salazar et al. [121] | 2012 | 2009 | TF | 30.00% | 130 | 420 | Peru | Latin America | Respondent driven sampling |
| Sotelo & Claudia [122] | 2011 | 2009 | TF | 34.00% | 152 | 441 | Argentina | Latin America | Unknown |
| Stephens, Bernstein, Philip et al. [123] | 2011 | 2006–2009 | TM | 2.90% | 7 | 69 | US | Global North | STI clinic visit |
| | | | TF | 11.21% | 25 | 223 | | | |

(*Continued*)

**Table 1.** (Continued)

| Authors | Year of publication | Year of data collection | Transgender sample | HIV prevalence (%) | HIV frequency (n) | Sample size | Country | Geographic region | Sampling method |
|---|---|---|---|---|---|---|---|---|---|
| Subramanian, Ramakrishnan, Aridoss et al. [124] | 2013 | 2005–2009 | TF | 12.00% | 48 | 404 | India | Asia | Cluster sampling |
| Toibaro, Ebensrtejin, Parlante et al. [125] | 2009 | 2002–2006 | TF | 27.60% | 29 | 105 | Argentina | Latin America | STI clinic visit |
| Van Veen, Götz, van Leeuwen et al. [126] | 2010 | 2002–2005 | TF | 19.00% | 13 | 69 | Netherlands | Global North | Cluster sampling |
| Waheed, Satti, Arshad et al. [127] | 2017 | 2015–2016 | TF | 16.40% | 22 | 134 | Pakistan | Asia | Convenience sampling |
| Wasantioopapokakorn, Manopaiboon, Phoorisri et al. [128] | 2018 | 2011–2016 | TF | 11.85% | 82 | 692 | Thailand | Asia | Convenience sampling |
| Weissman, Ngak, Srean et al. [129] | 2016 | 2012 | TF | 4.00% | 37 | 891 | Cambodia | Asia | Respondent driven sampling |
| World Health Organization [130] | 2016 | 2015–2016 | TF | 4.00% | 11 | 299 | Philippines | Asia | Surveillance |
| Wickersham, Gibson, Bazazi et al. [131] | 2017 | 2014 | TF | 12.00% | 24 | 193 | Malaysia | Asia | Respondent driven sampling |
| Zaccarelli, Spizzichino, Venezia et al. [132] | 2004 | 1993–2003 | TF | 31.50% | 149 | 473 | Italy | Global North | STI clinic visit |
| Zea, Reisen, del Rio-Gonzalez et al. [133] | 2015 | 2011 | TF | 13.79% | 8 | 58 | Colombia | Latin America | Respondent driven sampling |

TF = trans feminine; TM = trans masculine.

years of age in the countries from which we had prevalence data for trans feminine individuals, matched to year of data collection, was 21.5 (95% CI 6.3–73.7). In Latin America ($n = 7917$), standardized prevalence was 25.9% (95% CI 20.0% - 31.8%) and the overall OR for HIV infection, compared to individuals over 15 years of age, was 95.6 (95% CI 73.7–122.7). In Asia ($n = 14,798$), the standardized HIV prevalence was 13.5% (95% CI 2.3% - 17.7%) and the overall OR was 68.0 (95% CI 42.9–107.8). Lastly, in the Global North, thus in Australia, Europe, and North America ($n = 24,697$), the standardized HIV prevalence was 17.1% (95% CI 13.1% - 21.1%) and the overall OR for HIV infection was 48.4 (95% CI 28.2–83.9).

The standardized HIV prevalence by sampling method is reported in Table 5. The standardized HIV prevalence among transgender individuals when respondent driven sampling was employed (33 studies) was 23.3% (95% CI 18.0% - 28.4%). When prevalence rates were ascertained via STI clinic visits (26 studies), standardized prevalence was 17.4% (95% CI 12.2% - 22.7%). When convenience sampling was employed (14 studies), standardized prevalence was 19.7% (95% CI 14.8% - 24.5%) and when cluster sampling was employed (11 studies), the standardized prevalence was 19.6% (95% CI 14.4% - 24.9%). The remaining sampling methods were relatively infrequently employed (i.e., employed in 5 or fewer studies).

Next, we looked at the potential role of PrEP in reducing HIV prevalence among trans feminine individuals. Prior to the introduction of PrEP (1997–2011), the standardized HIV prevalence in US-based studies was 18.4% (95% CI 14.8% - 22.0%; Table 6) and the overall OR for HIV infection, compared to individuals over 15 years of age in the USA, was 53.5 (95% CI 29.7–96.5). After the introduction of PrEP (2012–2017), the standardized HIV prevalence in

**Table 2. Meta-analysis of HIV prevalence in trans feminine individuals compared to all adults (age 15+).**

| Country | Year of data collection | Number of samples | Sample size | Frequency of HIV among TF in the samples | Prevalence (95% CI)* | Odds Ratio (95%CI) | HIV prevalence in adults (95% CI) |
|---|---|---|---|---|---|---|---|
| Argentina | 2004 | 1 | 105 | 29 | 27.6 (19.1–36.2) | 147.2 (96.0–225.9) | 0.258 (0.257–0.260) |
| Argentina | 2007 | 2 | 385 | 131 | 34 (29.3–38.8) | 176 (142.6–217.4) | 0.292 (0.290–0.294) |
| Argentina | 2009 | 1 | 441 | 152 | 34.5 (30–38.9) | 168.3 (138.3–204.8) | 0.312 (0.310–0.314) |
| Australia | 2004 | 1 | 141 | 6 | 4.3 (0.9–7.6) | 44 (19.4–99.7) | 0.101 (0.099–0.102) |
| Australia | 2012 | 1 | 77 | 8 | 10.4 (3.6–17.2) | 92.1 (44.3–191.5) | 0.126 (0.124–0.127) |
| Bangladesh | 2012 | 1 | 889 | 7 | 0.8 (0.2–1.4) | 74.7 (35.5–157.3) | 0.011 (0.010–0.011) |
| Brazil | 1995 | 1 | 434 | 174 | 40.1 (35.5–44.7) | 304.8 (251.6–369.3) | 0.219 (0.218–0.220) |
| Brazil | 2002 | 1 | 122 | 24 | 19.7 (12.6–26.7) | 78 (49.9–121.9) | 0.313 (0.312–0.314) |
| Brazil | 2005 | 1 | 84 | 12 | 14.3 (6.8–21.8) | 46.7 (25.3–86.0) | 0.356 (0.355–0.357) |
| Brazil | 2006 | 1 | 284 | 71 | 25.0 (20.0–30.0) | 89.3 (68.2–116.8) | 0.372 (0.371–0.373) |
| Brazil | 2012 | 1 | 152 | 37 | 24.3 (17.5–31.2) | 69.7 (48.1–100.9) | 0.460 (0.459–0.461) |
| Brazil | 2015 | 1 | 345 | 141 | 40.9 (35.7–46.1) | 136.4 (110.1–169.1) | 0.504 (0.503–0.505) |
| Brazil | 2016 | 1 | 2846 | 843 | 29.6 (27.9–31.3) | 81.1 (74.9–87.9) | 0.516 (0.515–0.517) |
| Burkina Faso | 2013 | 1 | 108 | 3 | 2.8 (-0.3–5.9) | 2.7 (0.9–8.5) | 1.043 (1.036–1.049) |
| Cambodia | 2012 | 1 | 891 | 37 | 4.2 (2.8–5.5) | 5.7 (4.1–7.9) | 0.756 (0.751–0.762) |
| Cambodia | 2015 | 1 | 1375 | 81 | 5.9 (4.6–7.1) | 9.2 (7.3–11.5) | 0.678 (0.673–0.683) |
| China | 2016 | 1 | 498 | 38 | 7.6 (5.3–10) | 187.7 (134.8–261.3) | 0.044 (0.044–0.044) |
| Colombia | 2011 | 1 | 58 | 8 | 13.8 (4.9–22.7) | 40.2 (19.1–84.8) | 0.397 (0.394–0.399) |
| Côte d'Ivoire | 2015 | 1 | 298 | 76 | 25.5 (20.6–30.5) | 11.7 (9.1–15.2) | 2.832 (2.823–2.841) |
| El Salvador | 2008 | 1 | 67 | 13 | 19.4 (9.9–28.9) | 37.9 (20.7–69.4) | 0.631 (0.623–0.639) |
| India | 1997 | 1 | 84 | 38 | 45.2 (34.6–55.9) | 183.8 (119.6–282.4) | 0.447 (0.447–0.448) |
| India | 2006 | 1 | 575 | 104 | 18.1 (14.9–21.2) | 66.7 (53.9–82.5) | 0.330 (0.330–0.330) |
| India | 2007 | 2 | 529 | 71 | 13.4 (10.5–16.3) | 49.8 (38.8–63.9) | 0.310 (0.310–0.311) |
| India | 2011 | 1 | 176 | 27 | 15.3 (10.0–20.7) | 65.7 (43.6–99.0) | 0.275 (0.275–0.275) |
| India | 2012 | 1 | 82 | 11 | 13.4 (6.0–20.8) | 59.9 (31.7–113.0) | 0.258 (0.258–0.258) |
| India | 2009 | 1 | 51 | 21 | 41.2 (27.7–54.7) | 255.3 (146.1–445.8) | 0.273 (0.273–0.274) |
| Indonesia | 2002 | 1 | 241 | 53 | 22 (16.8–27.2) | 288.5 (212.7–391.4) | 0.098 (0.097–0.098) |
| Indonesia | 2007 | 2 | 1225 | 334 | 27.3 (24.8–29.8) | 160.4 (141.5–181.9) | 0.233 (0.232–0.234) |
| Indonesia | 2009 | 1 | 217 | 80 | 36.9 (30.4–43.3) | 214.1 (162.5–282.1) | 0.272 (0.271–0.273) |
| Indonesia | 2012 | 1 | 100 | 16 | 16.0 (8.8–23.2) | 61.5 (36.0–105.0) | 0.309 (0.308–0.310) |
| Italy | 1998 | 1 | 473 | 149 | 31.5 (27.3–35.7) | 390.4 (321.5–474.1) | 0.118 (0.117–0.119) |
| Italy | 2007 | 1 | 173 | 21 | 12.1 (7.3–17) | 65.7 (41.7–103.8) | 0.210 (0.208–0.211) |
| Italy | 2009 | 1 | 56 | 3 | 5.4 (-0.5–11.3) | 26.8 (8.4–85.6) | 0.211 (0.210–0.212) |
| Jamaica | 2015 | 1 | 103 | 26 | 25.2 (16.9–33.6) | 22.7 (14.5–35.4) | 1.468 (1.451–1.484) |
| Lebanon | 2012 | 1 | 40 | 4 | 10.0 (0.7–19.3) | 246.1 (87.5–692.4) | 0.045 (0.043–0.047) |
| Lesotho | 2013 | 1 | 71 | 42 | 59.2 (47.7–70.6) | 4.6 (2.9–7.4) | 23.950 (23.876–24.023) |
| Malawi | 2013 | 1 | 75 | 12 | 16.0 (7.7–24.3) | 1.8 (1.0–3.3) | 9.683 (9.664–9.703) |
| Malaysia | 2014 | 1 | 193 | 24 | 12.4 (7.8–17.1) | 39.9 (26.0–61.1) | 0.355 (0.352–0.357) |
| Mali | 2014 | 1 | 165 | 37 | 22.4 (16.1–28.8) | 220.9 (153.1–318.6) | 0.131 (0.128–0.133) |
| Mexico | 2012 | 1 | 100 | 22 | 22 (13.9–30.1) | 123.9 (77.2–198.9) | 0.227 (0.226–0.228) |
| Netherlands | 2003 | 1 | 69 | 13 | 18.8 (9.6–28.1) | 218.8 (119.7–400.2) | 0.106 (0.104–0.108) |
| Nigeria | 2014 | 1 | 105 | 75 | 71.4 (62.8–80.1) | 183 (119.8–279.4) | 1.348 (1.346–1.350) |
| Pakistan | 1998 | 1 | 208 | 0 | 0 (0.0–0.0) | 395.9 (24.6–6359.6) | 0.001 (0.001–0.001) |
| Pakistan | 2004 | 1 | 409 | 5 | 1.2 (0.2–2.3) | 451.5 (186.8–1091.6) | 0.003 (0.003–0.003) |
| Pakistan | 2005 | 1 | 1162 | 10 | 0.9 (0.3–1.4) | 73.0 (39.2–136.1) | 0.012 (0.012–0.012) |
| Pakistan | 2006 | 1 | 810 | 38 | 4.7 (3.2–6.1) | 243.0 (175.4–336.6) | 0.020 (0.020–0.021) |

*(Continued)*

**Table 2.** (Continued)

| Country | Year of data collection | Number of samples | Sample size | Frequency of HIV among TF in the samples | Prevalence (95% CI)* | Odds Ratio (95%CI) | HIV prevalence in adults (95% CI) |
|---|---|---|---|---|---|---|---|
| Pakistan | 2007 | 1 | 269 | 6 | 2.2 (0.5–4) | 78.4 (34.9–176.1) | 0.029 (0.029–0.029) |
| Pakistan | 2008 | 1 | 1181 | 75 | 6.4 (5–7.7) | 172.4 (136.4–217.9) | 0.039 (0.039–0.040) |
| Pakistan | 2009 | 1 | 306 | 66 | 21.6 (17–26.2) | 560.5 (426.8–736.1) | 0.049 (0.049–0.049) |
| Pakistan | 2015 | 1 | 134 | 22 | 16.4 (10.1–22.7) | 214.8 (136.0–339.2) | 0.091 (0.091–0.092) |
| Paraguay | 2011 | 1 | 237 | 64 | 27 (21.4–32.7) | 89.5 (67.2–119.3) | 0.411 (0.406–0.417) |
| Peru | 2007 | 1 | 214 | 64 | 29.9 (23.8–36) | 158.9 (118.5–212.9) | 0.268 (0.266–0.270) |
| Peru | 2008 | 1 | 208 | 35 | 16.8 (11.7–21.9) | 75.3 (52.4–108.3) | 0.268 (0.266–0.270) |
| Peru | 2009 | 1 | 420 | 130 | 31 (26.5–35.4) | 166.9 (135.7–205.3) | 0.268 (0.266–0.270) |
| Peru | 2011 | 1 | 713 | 104 | 14.6 (12–17.2) | 62.5 (50.7–76.9) | 0.273 (0.270–0.275) |
| Peru | 2013 | 1 | 89 | 30 | 33.7 (23.9–43.5) | 176.6 (113.8–274.1) | 0.287 (0.285–0.289) |
| Peru | 2014 | 1 | 310 | 61 | 19.7 (15.3–24.1) | 81.1 (61.3–107.4) | 0.301 (0.299–0.303) |
| Philippines | 2015 | 1 | 299 | 11 | 3.7 (1.5–5.8) | 50.9 (27.9–92.9) | 0.075 (0.074–0.076) |
| Senegal | 2013 | 1 | 199 | 74 | 37.2 (30.5–43.9) | 118.1 (88.6–157.4) | 0.499 (0.494–0.504) |
| Spain | 2000 | 1 | 60 | 14 | 23.3 (12.6–34) | 115.6 (63.5–210.2) | 0.263 (0.261–0.264) |
| Spain | 2005 | 1 | 529 | 129 | 24.4 (20.7–28) | 109.3 (89.6–133.3) | 0.294 (0.293–0.296) |
| Spain | 2008 | 1 | 142 | 18 | 12.7 (7.2–18.1) | 43.6 (26.6–71.5) | 0.332 (0.330–0.334) |
| Spain | 2010 | 1 | 101 | 46 | 45.5 (35.8–55.3) | 235.3 (159.1–348.1) | 0.354 (0.352–0.356) |
| Spain | 2015 | 1 | 453 | 40 | 8.8 (6.2–11.4) | 28.1 (20.3–38.8) | 0.344 (0.342–0.346) |
| Swaziland | 2013 | 1 | 120 | 17 | 14.2 (7.9–20.4) | 74.5 (44.6–124.4) | 0.221 (0.218–0.225) |
| Thailand | 2005 | 1 | 474 | 64 | 13.5 (10.4–16.6) | 12.8 (9.9–16.7) | 1.202 (1.199–1.205) |
| Thailand | 2008 | 1 | 140 | 13 | 9.3 (4.5–14.1) | 9 (5.1–16) | 1.121 (1.119–1.124) |
| Thailand | 2012 | 1 | 47 | 2 | 4.3 (-1.5–10) | 4.4 (1.1–18.1) | 1.002 (1.000–1.005) |
| Thailand | 2013 | 1 | 692 | 82 | 11.8 (9.4–14.3) | 13.7 (10.8–17.2) | 0.975 (0.973–0.978) |
| Thailand | 2015 | 2 | 1221 | 108 | 8.8 (7.3–10.4) | 10.6 (8.7–12.9) | 0.908 (0.906–0.911) |
| Togo | 2013 | 1 | 51 | 9 | 17.6 (7.2–28.1) | 8.9 (4.3–18.2) | 2.359 (2.345–2.374) |
| Uruguay | 1999 | 1 | 200 | 49 | 24.5 (18.5–30.5) | 111.3 (80.6–153.8) | 0.291 (0.284–0.297) |
| US | 1997 | 1 | 392 | 137 | 34.9 (30.2–39.7) | 190.7 (154.9–234.7) | 0.281 (0.280–0.282) |
| US | 1998 | 2 | 482 | 91 | 18.9 (15.4–22.4) | 81.0 (64.5–101.7) | 0.287 (0.286–0.287) |
| US | 2000 | 1 | 538 | 161 | 29.9 (26.1–33.8) | 138.9 (115.5–167.1) | 0.306 (0.306–0.307) |
| US | 2004 | 2 | 141 | 30 | 21.3 (14.5–28) | 82.3 (55.0–123.2) | 0.327 (0.327–0.328) |
| US | 2005 | 3 | 852 | 81 | 9.5 (7.5–11.5) | 31.1 (24.8–39.2) | 0.336 (0.335–0.337) |
| US | 2007 | 1 | 223 | 25 | 11.2 (7.1–15.4) | 37.4 (24.7–56.7) | 0.337 (0.336–0.337) |
| US | 2009 | 2 | 602 | 195 | 32.4 (28.7–36.1) | 134.8 (113.7–159.9) | 0.354 (0.353–0.355) |
| US | 2010 | 3 | 16943 | 664 | 3.9 (3.6–4.2) | 10.8 (10–11.7) | 0.375 (0.374–0.376) |
| US | 2011 | 2 | 657 | 75 | 11.4 (9–13.8) | 33.5 (26.3–42.6) | 0.383 (0.383–0.384) |
| US | 2012 | 1 | 292 | 13 | 4.5 (2.1–6.8) | 11.8 (6.8–20.7) | 0.392 (0.391–0.392) |
| US | 2013 | 3 | 729 | 197 | 27 (23.8–30.2) | 94.1 (79.9–110.8) | 0.392 (0.391–0.393) |
| US | 2016 | 2 | 380 | 148 | 38.9 (34–43.9) | 166.6 (135.5–204.7) | 0.382 (0.381–0.382) |
| US | 2003 | 1 | 192 | 16 | 8.3 (4.4–12.2) | 18.9 (11.3–31.5) | 0.479 (0.478–0.480) |
| Vietnam | 2004 | 1 | 75 | 5 | 6.7 (1–12.3) | 24.7 (10–61.1) | 0.289 (0.287–0.290) |
| Vietnam | 2015 | 1 | 205 | 37 | 18 (12.8–23.3) | 68.4 (47.9–97.6) | 0.321 (0.320–0.322) |
| Overall | - | - | - | - | 19.9 (14.7–25.1)* | 66.0 (51.4–84.8) | - |

Note. Heterogeneity: $Q = 6327.25$, $df = 86$, $p < .0001$, $I^2 = 98.63\%$.

*Overall prevalence was calculated by direct standardization based on country-year weights used in meta-analysis.

**Table 3. Meta-analysis of HIV prevalence in trans masculine individuals compared to all adults (age 15+).**

| Country | Year of data collection | Number of samples | Sample size | Frequency of HIV among TM in the samples | Prevalence (95% CI)* | Odds Ratio (95% CI) | HIV prevalence in adults (95% CI) |
|---|---|---|---|---|---|---|---|
| Australia | 2004 | 1 | 17 | 0 | 0.0 (0.0–0.2) | 28.3 (1.7–470.5) | 0.101 (0.099–0.102) |
| Australia | 2012 | 1 | 28 | 1 | 3.6 (-3.3–10.4) | 29.4 (4–216.5) | 0.126 (0.124–0.127) |
| Brazil | 2002 | 1 | 16 | 0 | 0.0 (0.0–0.2) | 9.7 (0.6–160.9) | 0.313 (0.312–0.314) |
| Brazil | 2006 | 1 | 51 | 0 | 0.0 (0.0–0.1) | 2.6 (0.2–42.1) | 0.372 (0.371–0.373) |
| Canada | 2013 | 1 | 11 | 0 | 0.0 (0.0–0.3) | 19.1 (1.1–323.8) | 0.227 (0.226–0.229) |
| Italy | 2007 | 1 | 20 | 0 | 0.0 (0.0–0.2) | 11.6 (0.7–191.9) | 0.210 (0.208–0.211) |
| Italy | 2009 | 1 | 27 | 0 | 0.0 (0.0–0.1) | 8.6 (0.5–140.9) | 0.211 (0.210–0.212) |
| Spain | 2008 | 1 | 92 | 2 | 2.2 (-0.8–5.2) | 6.7 (1.6–27.1) | 0.332 (0.330–0.334) |
| US | 1997 | 1 | 123 | 2 | 1.6 (-0.6–3.9) | 5.9 (1.5–23.7) | 0.281 (0.280–0.282) |
| US | 2003 | 1 | 50 | 0 | 0.0 (0.0–0.1) | 3.0 (0.2–48.3) | 0.331 (0.330–0.332) |
| US | 2005 | 2 | 124 | 2 | 1.6 (-0.6–3.8) | 4.7 (1.2–19.2) | 0.345 (0.344–0.345) |
| US | 2007 | 2 | 92 | 8 | 8.7 (2.9–14.5) | 26.2 (12.7–54.2) | 0.362 (0.361–0.362) |
| US | 2009 | 2 | 292 | 2 | 0.7 (-0.3–1.6) | 1.8 (0.5–7.3) | 0.379 (0.378–0.379) |
| US | 2010 | 2 | 5256 | 21 | 0.4 (0.2–0.6) | 1.0 (0.7–1.6) | 0.387 (0.386–0.388) |
| US | 2011 | 1 | 120 | 10 | 8.3 (3.4–13.3) | 22.9 (12–43.8) | 0.395 (0.395–0.396) |
| US | 2012 | 1 | 30 | 1 | 3.3 (-3.1–9.8) | 8.7 (1.2–63.7) | 0.396 (0.395–0.396) |
| US | 2013 | 1 | 122 | 0 | 0.0 (0.0–2.8) | 1.0 (0.1–16.7) | 0.392 (0.391–0.393) |
| Overall | - | - | - | - | 2.56 (0.0–5.9)* | 6.8 (3.6–13.1) | |

Note. Heterogeneity: $Q = 13.06$, $df = 16$, $p<0.0001$, $I^2 = 70.62\%$.

*Overall prevalence was calculated by direct standardization based on country-year weights used in meta-analysis.

US-based studies was 23.7% (95% CI 20.2% - 27.2%) and the OR for HIV infection, compared to individuals over 15 years of age, was 58.0 (95% CI 12.3–275.9). The forest plot of this analysis is presented in Fig 4.

Heterogeneity was high across the studies that included trans feminine individuals ($Q = 6327.25$, $df = 86$, $p < .0001$, $I^2 = 98.63\%$). This may be because the studies were conducted in different countries using different methodologies. Heterogeneity was moderate for the studies that included trans masculine individuals ($Q = 102.06$, $df = 15$, $p<0.0001$, $I^2 = 72.17\%$). The funnel plots showed an asymmetrical distribution of studies and may therefore indicate publication bias (S2 Appendix in Fig A2.1 and Fig A2.2).

## Discussion

This systematic review and meta-analysis affirms that transgender individuals are disproportionately burdened by HIV, and that this is the case for not only trans feminine individuals, but also for trans masculine individuals. Using a larger pooled sample than ever compiled before, we ascertained that trans masculine individuals almost seven times more likely to have HIV, and trans feminine individuals are 66 times more likely to have HIV, than other individuals over 15 years of age. Additionally, based on data from 34 countries across major geographic regions, we found support for the contention that the disproportionate burden for HIV carried by transgender individuals is a worldwide phenomenon, and that some regions, such as Africa and Latin America, may be impacted more than others. Further, we established that sampling methods are likely to impact prevalence rates and that, to date, PrEP prevention effects on HIV prevalence cannot be established.

To our knowledge, no previous study has estimated the HIV burden carried by trans masculine individuals worldwide. Reisner and Murchinson [1] did conduct a research synthesis of

Country, Year

Log Odds Ratio [95% CI]

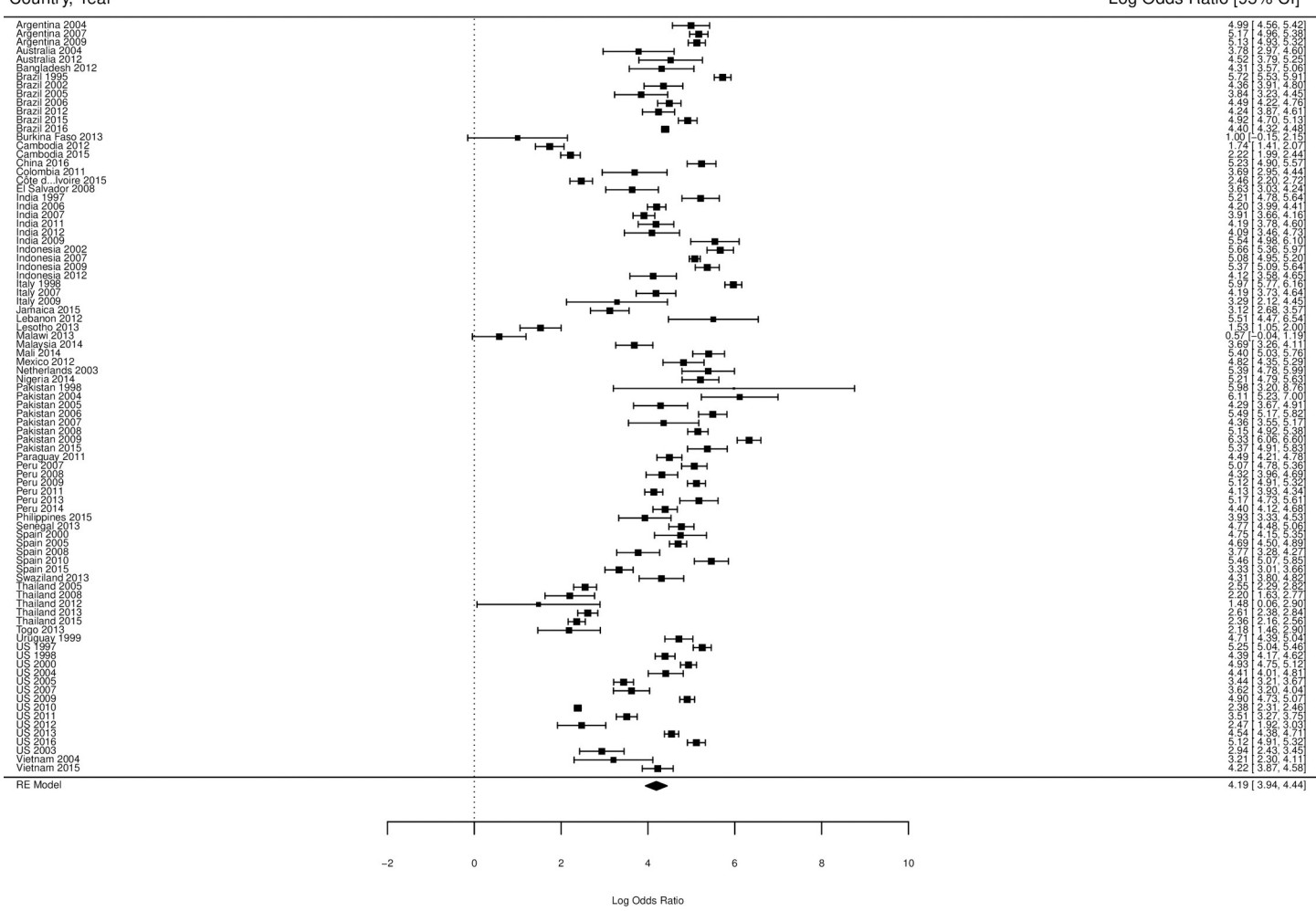

**Fig 2. Forest plot of HIV prevalence in trans feminine individuals compared to all adults (age 15+).** The scale on the x-axis is log odds ratio. The percentages indicate the weight of each country by year within the meta-analysis. The numbers in the right column are the log odds ratios including their confidence intervals. We converted these log odds ratios into odds ratios, as described in *Table 2*.

HIV risks in trans masculine individuals where laboratory-confirmed prevalence ranged from 0% to 4.3% and Becasen and colleagues [17] established a laboratory-confirmed estimated prevalence rate of 3.2% but, in both studies, no odds ratios were calculated to ascertain the relative burden of HIV carried by trans masculine individuals. Our finding that trans masculine individuals are almost seven times more likely to have HIV than other individuals over 15 years of age indicates that many trans masculine individuals are indeed at risk for HIV. The presumption that trans masculine individuals almost exclusively have sex with cis-gender women and are therefore not at risk for HIV is thus incorrect [1]. As indicated by Reisner and Murchinson, there is a diverse range of bio-anatomies represented among trans masculine individuals and their partners in sexual encounters, and these should be considered in HIV prevention efforts [1].

Our finding that trans feminine individuals are 66 times more likely to have HIV than other individuals over 15 years of age is a higher estimate that the estimate generated in Baral and colleagues' meta-analysis, [4] where the odds ratio for HIV infection among transgender women was 49.1. We believe that the odds ratio and prevalence rates established in our

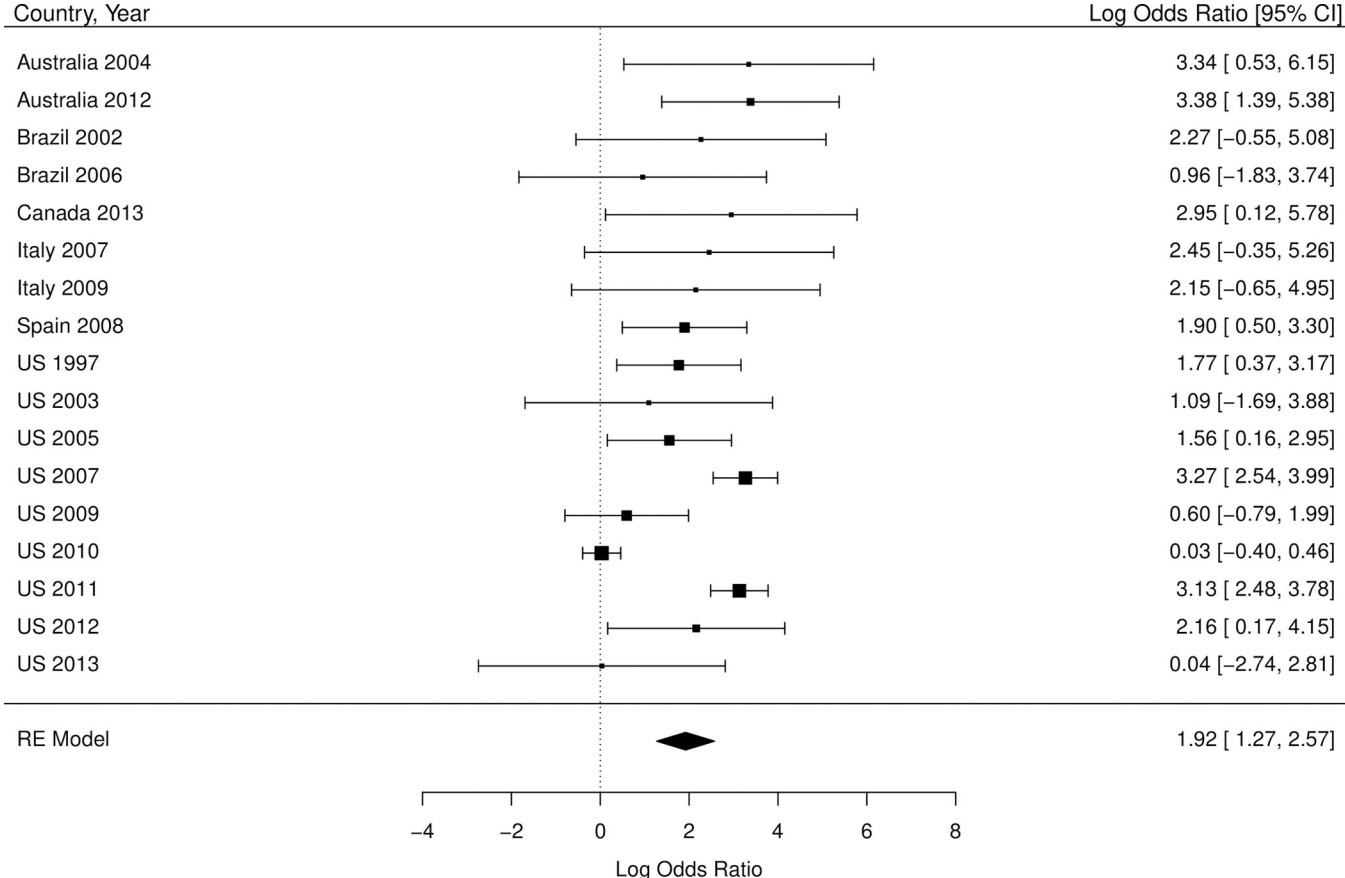

**Fig 3. Forest plot of HIV prevalence in trans masculine individuals compared to all adults (age 15+).** The scale on the x-axis is log odds ratio. The percentages indicate the weight of each country by year within the meta-analysis. The numbers in the right column are the log odds ratios including their confidence intervals. We converted these log odds ratios into odds ratios, as described in *Table 3*.

systematic review and meta-analysis are likely more realistic estimations for two reasons. First, our methodological approach used standardized rather than pooled prevalence and took into account not only country but also year of data collection. In a pooled estimate, the total study population and total HIV cases are summed, and then a crude proportion is calculated. This does not take heterogeneity and variation among the included studies into account. Our standardization approach entailed taking the weights from each country-year into account. Without the weighted standardization, a country-year combination that contains large or small

**Table 4. HIV prevalence and odds ratios for trans feminine individuals compared to all adults (age 15+), separated by geographic region.**

| Region | Number of countries | Number of Samples | Sample size | Prevalence (95% CI) * | Odds Ratio (95% CI) * | HIV prevalence in adults (95% CI)* |
|---|---|---|---|---|---|---|
| Africa | 9 | 9 | 1192 | 29.9 (22.5–37.3) | 21.5 (6.3–73.7) | 4.69 (4.67–4.71) |
| Asia | 11 | 35 | 14798 | 13.5 (2.3–17.7) | 68.0 (42.9–107.8) | 0.344 (0.343–0.345) |
| Global North | 5 | 35 | 24697 | 17.1 (13.1–21.1) | 48.4 (28.2–83.9) | 0.297 (0.296–0.298) |
| Latin America | 9 | 23 | 7917 | 25.9 (20.0–31.8) | 95.6 (73.7–122.7) | 0.391 (0.388–0.394) |

Note. The HIV prevalence in adults of the population (last column) is the weighted prevalence of the countries included in this meta-analysis, not overall prevalence in the region.

* Results were calculated by direct standardisation of country-year sample size instead of pooling.

**Table 5. HIV prevalence in trans feminine individuals, separated by sampling method.**

| Sampling method | Number of samples | Sample size | HIV prevalence (95% CI)* |
|---|---|---|---|
| Respondent driven sampling | 33 | 12202 | 23.3 (18.0–28.4) |
| STI clinic visit | 26 | 19360 | 17.4 (12.2–22.7) |
| Convenience sampling | 14 | 3733 | 19.7 (14.8–24.5) |
| Cluster sampling | 11 | 4273 | 19.6 (14.4–24.9) |
| Hospital | 5 | 827 | 15.0 (9.8–20.4) |
| Snowball | 4 | 1203 | 11.8 (8.0–15.6) |
| Surveillance | 4 | 2339 | 9.1 (6.1–12.0) |
| NGO | 2 | 213 | 37.8 (31.5–44.2) |
| Database health plan | 1 | 3475 | 5.4 (4.6–6.1) |
| Purposive sampling | 1 | 538 | 29.9 (26.1–33.8) |

Note. For one study, the sampling method was unknown and is not included in this table.

*: Results were calculated by direct standardisation of country-year sample size instead of pooling.

study samples is likely to deliver misleading pooled results. The standardized approach thus delivers a more robust estimation than a pooled approach. Second, due to recent increases in the number of studies reporting HIV prevalence among transgender individuals, the total sample of transgender individuals in our meta-analysis was almost four and a half times larger than the pooled sample in Baral et al. [4] Third, the data reviewed in Baral et al. was derived from 15 countries, all of which have male-dominant epidemics, while the data in the meta-analysis reported here was derived from 34 countries, thus lending additional support to the contention that the high burden of HIV among transgender individuals is a worldwide phenomenon.

Our finding that HIV prevalence among transgender individuals appears to be, over the course of the epidemic, higher in African and Latin American regions may point to greater disapproval of gender fluidity and the accompanying marginalization that puts transgender individuals more at risk for HIV in these regions, although we recognize that overall prevalence rates for HIV are higher in Sub-Saharan Africa than in many other regions. Nonetheless, this was the first systematic review and meta-analysis to include samples from Sub-Saharan Africa, and the findings from Sub-Saharan Africa point to a significant burden of HIV among transgender individuals. However, given that, in our analyses, the sample sizes for African regions and Latin America were smaller than the sample sizes for other regions, more research is needed to confirm that transgender individuals in these regions do indeed have higher prevalence rates and carry an even greater burden of HIV. Additionally, future research should also seek to establish HIV prevalence rates and burdens in other understudied regions such as Eastern Europe.

This meta-analysis also demonstrated that sampling methods are likely to impact prevalence rates. This is in line with critiques of sampling methods that were levied in earlier

**Table 6. HIV prevalence and odds ratios for trans feminine individuals compared to all adults (age 15+) in US-based studies, according to whether data was collected before or after the introduction of PrEP (2012).**

| | Number of studies | Sample size | Frequency of HIV among TF in the samples | Prevalence (95%CI)* | Odds Ratio (95%CI) |
|---|---|---|---|---|---|
| Before PrEP | 18 | 21022 | 1475 | 18.4 (14.8–22.0) | 53.5 (29.7–96.5) |
| After PrEP | 6 | 1401 | 358 | 23.7 (20.2–27.2) | 58.0 (12.3–275.9) |

Note. *Overall prevalence was calculated by direct standardization based on country-year weights used in meta-analysis.

## Before the introduction of PrEP

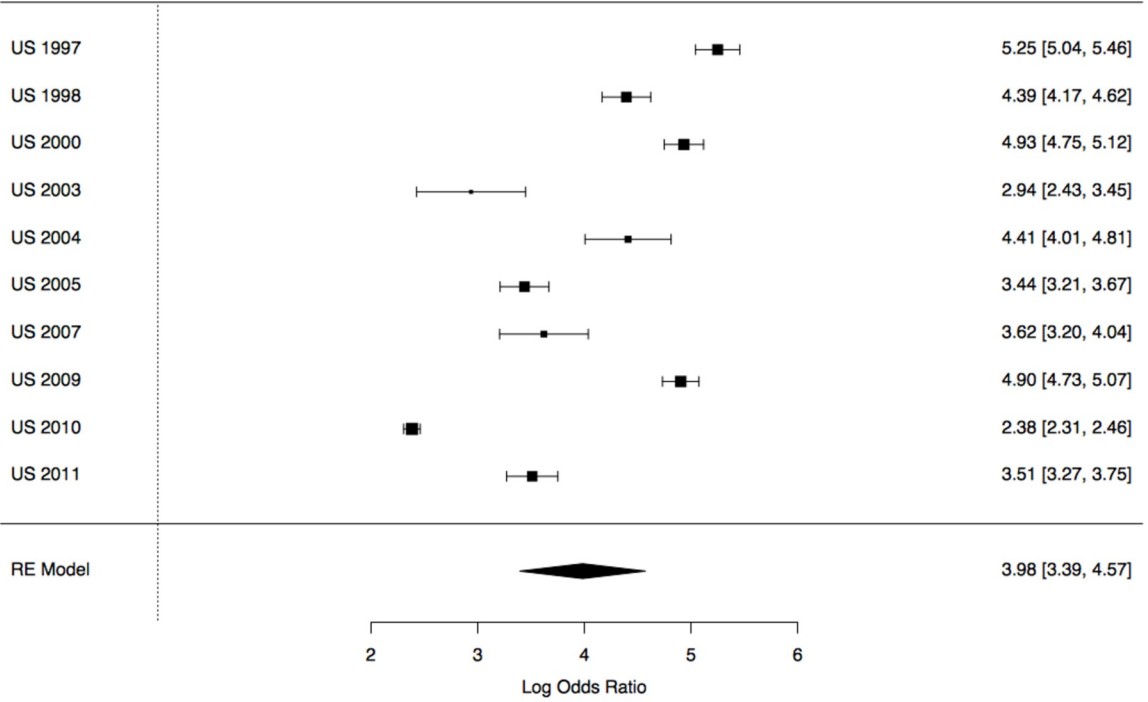

## After the introduction of PrEP

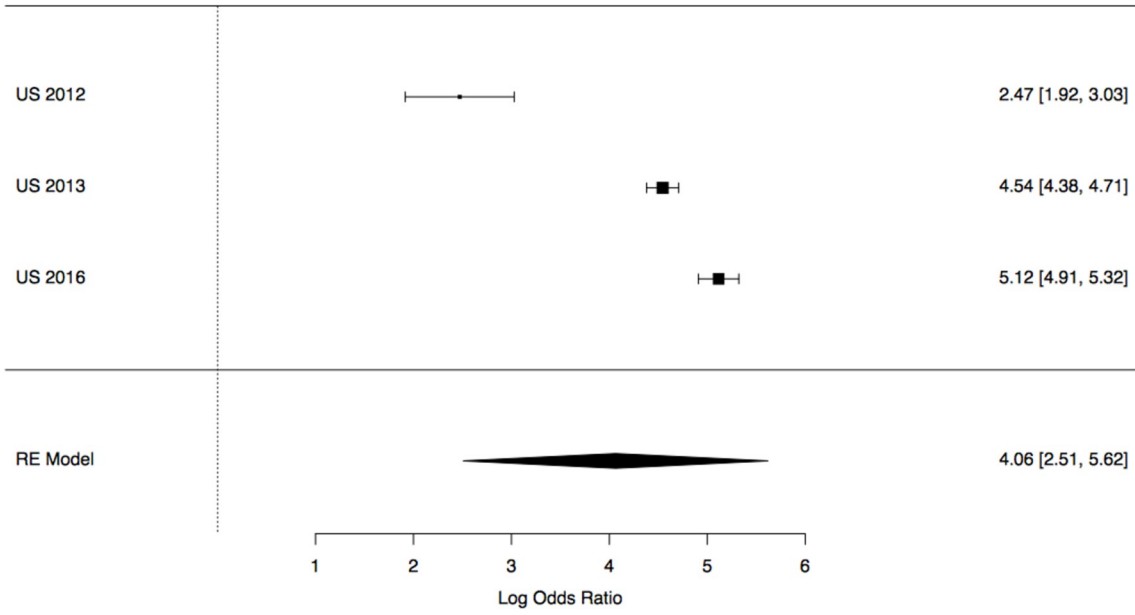

**Fig 4. Forest plot of HIV prevalence in trans feminine individuals in the USA compared to all adults (age 15+) in the USA.** The 10 country-year including 18 studies above the line are studies where data were collected prior to the introduction of PrEP (2012). The 3 country-year including 6 studies below the line are studies where data were collected after the introduction of PrEP. The scale on the x-axis is log odds ratio. The percentages indicate the weight of each sample within the meta-analysis. The numbers in the right column are the log odds ratios including their confidence intervals.

commentaries on Baral et al. [4] and in other studies [11, 18, 134]. In our study, the various sampling methods generated very different prevalence rates for HIV in trans feminine individuals ranging from 5.4% to 37.8%. However, the four most frequently used sampling methods, namely respondent-driven sampling, sampling via STI clinics, convenience sampling, and cluster sampling had similar ranges (17.4% to 23.3%). We believe that the impact of sampling methods on prevalence rates is in need of further investigation. In our analyses, unambiguous classification was not always possible and the prevalence rates generated for less common sampling methods may be less reliable. We therefore recommend more comprehensive investigations of the impact of sampling methods in transgender studies.

In our meta-analyses, we also explored the potential role of PrEP availability by comparing studies conducted in the US where data was collected prior to and after the introduction of PrEP. No effect of PrEP could be established yet in our analyses. In fact, we found a higher HIV prevalence rate following the introduction of PrEP. This may be because there were only six studies done following the introduction of PrEP and the total sample after PrEP introduction was smaller and possibly less representative than the 18 studies conducted before PrEP was introduced. It is possible that no reduction in prevalence due to PrEP is the result of PrEP not yet reaching trans individuals. The inclusion of transgender individuals in PrEP trials has been low and access to PrEP for transgender individuals has been limited.[20, 21, 135] However, a qualitative study on PrEP acceptability among transgender women in San Francisco showed that interest was relatively high once participants were informed about the possibilities, thus suggesting that transgender individuals at high risk for HIV need to be informed about PrEP [20]. By the same bio-medical token, future meta-analytic studies should also include Treatment-as-prevention (TasP) effects in their analysis, once sufficiently robust primary data is available.

This systematic review and meta-analysis should be interpreted in light of possible limitations. One is potential sample size biases for studies originating from countries other than the USA, and those of trans masculine individuals. To be able to present a comprehensive, global picture, we set a lower bound for trans feminine individuals, excluding sample sizes of trans feminine individuals less than 40. Yet, we did not apply a minimum sample size for studies among trans masculine individuals as this would have resulted in the exclusion of most studies reporting HIV prevalence among trans masculine individuals. Further, we were not in a position to conduct city-level comparisons and thus acknowledge that our country-level analyses may provide a less precise estimation of the odds ratios, as these do not take into account that, in some countries, the HIV epidemic is more concentrated in certain areas. A third possible limitation is related to our classification of sampling methods. Unambiguous classification was not always feasible and it is possible that the prevalence rates generated for less common sampling methods were less reliable. Fourth, in our meta-analysis, the sample sizes for African regions and Latin America were smaller than the sample sizes for other regions, and this may have impacted the prevalence rates. Additionally, no prevalence rates from Eastern Europe were available. Fifth, our analysis did not account for sexual orientation or the presence or absence of gender reassignment surgery, both of which can impact HIV risk. It is also did not separately ascertain prevalence rates for trans feminine individuals who engage in sex work versus those who do not as primary level data on this is not available on a global scale. We recommend that future research take these potential shortcomings into consideration. Specifically, we recommend that future research explicitly investigate prevalence among sub-populations within the transgender community, and that new studies also take changes in HIV treatment (TasP) and sampling strategies, as well as their interactions, into account, as this will provide an even more comprehensive picture of HIV prevalence and burden among transgender individuals.

## Summary and recommendations

That transgender individuals, both trans feminine and trans masculine, are, worldwide, disproportionately burdened by HIV points to the need to pay explicit attention to the unique HIV prevention and care needs of transgender individuals. HIV surveillance and research has traditionally grouped transgender individuals, particularly trans feminine individuals, with men who have sex with men (MSM), thus conflating gender with anatomy. This obscures the unique situation and vulnerabilities to HIV of transgender people [100]. It is therefore necessary to abandon the aggregation of data across MSM and trans feminine women [100, 136]. Additionally, and in line with MacCarthy et al., [11] we also propose disaggregating data across trans feminine and trans masculine individuals [106]. Although some individual and structural risk factors for HIV may be shared by transgender individuals, trans feminine and trans masculine individuals have unique needs [106].

Ascertaining that transgender individuals continue to be disproportionately burdened by HIV is important as it can serve as an impetus for efforts to change this burden. Although transgender individuals in certain regions may more affected by HIV and knowing that transgender processes are diverse across the world, we contend that it is important to, in all regions, target multiple levels of HIV risk, as well as their antecedents and their intersections, while being cognizant of the local context. Targeting individual level risk factors, such as unprotected sex, STI co-infection, and needle sharing, must occur alongside broader efforts to support transgender individuals and reduce stigmatization and marginalization [9, 137, 138].

Paramount to HIV risk reduction is gender affirmation, and in the context of HIV, gender affirmation is particularly important in health care [139–141]. Discrimination, judgment, insensitivity, and a lack of understanding from health care providers prevents many transgender individuals from accessing HIV prevention, testing, treatment, and care services [142]. Gender affirming care is not simply the provision of hormones and gender-affirming surgeries; it also includes using patients' preferred names and pronouns, respecting diversity in gender identities and expressions, employing inclusive intake forms, displaying images that are welcoming to transgender individuals, and creating safe spaces where transgender individuals can be themselves [135, 143]. We recommend integrating HIV prevention and care services in broader gender-affirming care services [6, 135, 142]. This includes actively making PrEP available to transgender individuals [141, 144, 145].

## Conclusion

In sum, this systematic review and meta-analyses have served to update our understanding of HIV prevalence over the course of the epidemic as well as HIV burden in both trans feminine and trans masculine individuals using a larger sample than ever before, and has shown that, worldwide, both carry a substantially higher burden of HIV than other individuals over 15 years of age. It has further demonstrated that a by country and year analysis is recommended, that prevalence rates are higher in African and Latin American regions, that sampling methods may impact prevalence rates, and that, at this point in time, the evidence does not suggest that PrEP has played a role in reducing HIV among transgender individuals.

## Supporting information

**S1 Appendix. Initial meta-analyses.**
(DOCX)

**S2 Appendix.** Fig A2.1. Funnel plot for the countries describing data from trans feminine individuals. Fig A2.2. Funnel plot for the countries describing data from trans masculine

individuals.
(ZIP)

**S1 Checklist. PRISMA 2009 checklist.**
(DOC)

## Acknowledgments

We thank Kristopher Banham (Cardiff University) and Ngoc Phuong Trinh Nguyen (Maastricht University) for their assistance with data extraction.

## Author Contributions

**Conceptualization:** Sarah E. Stutterheim, Mart van Dijk, Kai J. Jonas.

**Data curation:** Mart van Dijk, Haoyi Wang.

**Formal analysis:** Sarah E. Stutterheim, Mart van Dijk, Haoyi Wang, Kai J. Jonas.

**Funding acquisition:** Sarah E. Stutterheim, Kai J. Jonas.

**Methodology:** Sarah E. Stutterheim, Mart van Dijk, Haoyi Wang, Kai J. Jonas.

**Project administration:** Sarah E. Stutterheim.

**Supervision:** Sarah E. Stutterheim, Kai J. Jonas.

**Validation:** Mart van Dijk, Haoyi Wang.

**Visualization:** Mart van Dijk, Haoyi Wang.

**Writing – original draft:** Sarah E. Stutterheim, Mart van Dijk, Kai J. Jonas.

**Writing – review & editing:** Sarah E. Stutterheim, Mart van Dijk, Haoyi Wang, Kai J. Jonas.

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
