## [Decision Letter · Decision Letter 0]

16 Apr 2020

PONE-D-20-01449

The worldwide burden of HIV in transgender individuals: An updated systematic review and meta-analysis

PLOS ONE

Dear Dr. Stutterheim,

Thank you for submitting your manuscript to PLOS ONE. After careful consideration, we feel that it has merit but does not fully meet PLOS ONE’s publication criteria as it currently stands. Therefore, we invite you to submit a revised version of the manuscript that addresses the points raised during the review process.

We would appreciate receiving your revised manuscript by May 31 2020 11:59PM. To enhance the reproducibility of your results, we recommend that if applicable you deposit your laboratory protocols in protocols.io, where a protocol can be assigned its own identifier (DOI) such that it can be cited independently in the future. For instructions see: http://journals.plos.org/plosone/s/submission-guidelines#loc-laboratory-protocols

We look forward to receiving your revised manuscript.

Kind regards,

Chongyi Wei, DrPH

Academic Editor

PLOS ONE

Journal Requirements:

2. Thank you for your submission to PLOS ONE. We note that your literature search was performed on January 2019;to allow an up-to-date view of the topic, we would request that the search is updated.Moreover, we would suggest to  include the Funnel Plots (shown as Supplementary material) as a main Figure. Finally, we suggest that you report more fully the results of your quality assessment, indicating how each included study scored on every item of the scale.

Reviewers' comments:

Reviewer's Responses to Questions

**Comments to the Author**

1. Is the manuscript technically sound, and do the data support the conclusions?

Reviewer #1: Yes

Reviewer #2: Yes

2. Has the statistical analysis been performed appropriately and rigorously? 

Reviewer #1: Yes

Reviewer #2: Yes

3. Have the authors made all data underlying the findings in their manuscript fully available?

Reviewer #1: Yes

Reviewer #2: Yes

4. Is the manuscript presented in an intelligible fashion and written in standard English?

Reviewer #1: Yes

Reviewer #2: Yes

5. Review Comments to the Author

Reviewer #1: The authors have provided an ambitious and exhaustive systematic review and meta-analysis of laboratory-confirmed HIV prevalence rates among trans individuals, worldwide, from studies published between 2000-2019. The search and analytic methods are generally sound, the data are valuable, and the conclusions are generally well-considered. In general terms, the key concern with this manuscript is the wide time range in the review sampling frame, which brings up certain issues needing further consideration and clarification. These include:

1) Given the number of systematic reviews and meta-analyses on HIV prevalence in this population, including within the last 5 years, additional rationale for embarking on this study could be presented in the Introduction. What questions are the authors asking and answering that previous work has not covered, and why is it important? This is especially relevant given that the search time frame goes back so far, and prior studies have incorporated much of the pre-2015 studies included here.

2) The use of a country-specific, general population HIV prevalence rate to make comparisons and generate odds ratios is an innovative approach, but it requires greater clarification and specificity in the Methods. The vast majority of the studies included are not country-level (or even city-level), but rather much more local. This makes the comparison group somewhat spurious, as the cities wherein trans people's HIV prevalence was assessed will in almost all cases have higher underlying HIV prevalence rates in the general population than will each respective country as a whole. Furthermore, it is unclear whether the country-level HIV prevalence rates are aligned with the sampling year(s) of each respective study: if they are not (and I can't tell here), then they probably need to be readjusted to the respective study year before generating OR.

3) Please consider using study year as a moderator of HIV prevalence throughout. I appreciate the example shown with PrEP in U.S.-based studies; but the rollout of ART, PEPFAR, and subsequent treatment-as-prevention policies likely have a greater effect on HIV prevalence than PrEP does, including among trans people. It would be helpful to sampling year(s) as a moderating variable throughout these analyses, for instance using 5-year periods (2000-2004; 2005-2009, etc) to categorize moderating effects. Without this, you cannot show trends in HIV infection in this population as clearly; and more importantly, it is hard to tell where we are now (compared to where we were 20 years ago).

4) Typo "indivi" on p. 25, line 121.

Thank you for the opportunity to review this comprehensive and important manuscript.

Reviewer #2: I reviewed the statistical approach used in this paper only. The approach that was used is suitable, with one caveat: In Table 3, sample sizes from countries outside the US appear too small to estimate prevalence. Can the authors comment on the acceptability of using such small samples?

6. PLOS authors have the option to publish the peer review history of their article (what does this mean?). If published, this will include your full peer review and any attached files.

Reviewer #1: No

Reviewer #2: No

---

## [Author Response · Author response to Decision Letter 0]

5 May 2020

Reviewer 1

1) Given the number of systematic reviews and meta-analyses on HIV prevalence in this population, including within the last 5 years, additional rationale for embarking on this study could be presented in the Introduction. What questions are the authors asking and answering that previous work has not covered, and why is it important? This is especially relevant given that the search time frame goes back so far, and prior studies have incorporated much of the pre-2015 studies included here.

Thank you for this comment. The introduction in the revised manuscript outlines why a comprehensive, updated systematic review and meta-analysis is required. We argue that an updated review is needed because HIV risk is dynamic, particularly for transgender individuals (pg. 4). We also outline limitations of previous reviews, particularly those that have been published more recently. For example, Reisner and Murchison (2016) focused only on HIV prevalence in transmasculine individuals. Our systematic review and meta-analysis focuses on both transmasculine and transfeminine individuals. Poteat and colleagues’s (2016) article comprised only a systematic review and not a meta-analysis, and the same applies for MacCarthy et al (2017). Furthermore, MacCarthy et al. provide only very limited insight on HIV prevalence as the inclusion criteria for their systematic review required that studies reported both HIV and STI prevalence, which yielded only six studies. Lastly, Becasen and colleagues (2019) systematic review and meta-analysis focused only on HIV prevalence in the US while our looks at the worldwide burden of HIV infection. Our systematic review and meta-analyses is therefore more comprehensive that other reviews published since Baral et al. (2013). We make this explicit on page 6 of the revised manuscript. Also, we outline that our updated systematic review and meta-analysis further adds to the literature by exploring sampling frames and the potential impact of PrEP (pg. 6).

With regards to the time frame chosen, we followed Baral et al. (2013) in the our approach to this systematic review and meta-analysis as we felt this would enable clearer comparison between what was ascertained then and what we ascertained in our updated review. We considered this to be more informative than choosing a time frame after Baral et al’s publication. In fact, we believe that the longer time frame provides a more comprehensive and nuanced picture of HIV prevalence and burden for transgender individuals. We now state this explicitly under ‘Search strategies and eligibility’ on page 7. 

2) The use of a country-specific, general population HIV prevalence rate to make comparisons and generate odds ratios is an innovative approach, but it requires greater clarification and specificity in the Methods. The vast majority of the studies included are not country-level (or even city-level), but rather much more local. This makes the comparison group somewhat spurious, as the cities wherein trans people's HIV prevalence was assessed will in almost all cases have higher underlying HIV prevalence rates in the general population than will each respective country as a whole. Furthermore, it is unclear whether the country-level HIV prevalence rates are aligned with the sampling year(s) of each respective study: if they are not (and I can't tell here), then they probably need to be readjusted to the respective study year before generating OR.

Thank you for this comment. We acknowledge that using country-specific, general population HIV prevalence to make comparisons and generate odd ratios may be somewhat limited in terms of precision, particularly in contexts where city-level rates differ from country-level rates. While we considered the possibility of presenting the data at the city-level instead of country-level, we choose not to do this. We felt that that it would be more parsimonious to present the data at a country level. Country level data are, in our opinion, more useful as this data allowed for the same calculation for every country, thus increasing the comparability between countries. The aim of a meta-analysis is to provide a general overview, and presenting data at a city-level would have added substantial additional complexity that would undermine the ultimate aim of the meta-analysis. Furthermore, city-level data were often not available (many countries collect data nationally rather than per city) and there were many studies included that derived prevalence rates across various cities within a country. Additionally, for most countries, we combined multiple studies from several cities, thus further justifying our country-level approach. 

This country-level approach has previously been employed and we now explicate this on page 10 of the revised manuscript, referring to three prior meta-analyses that used this approach. We also immediately acknowledge, in the text, the limitations of this approach. To wit: “we acknowledge that this approach may provide a less precise estimation of the odds ratios, as it does not take into account that, in some countries, the HIV epidemic is more concentrated in certain areas.” (pg, 10). 

Thanks also for pointing out that we neglected to state the year upon which we based country-level HIV prevalence rates. We now make explicit that this was 2017 on page 10 of the revised manuscript. We did not adjust these rates based on sampling years of the studies for two reasons. Firstly, in most countries, HIV prevalence in the general population has been quite stable over time. As a result, we do not expect that adjustment based on sampling year would have a large effect. Secondly, we believe that this adjustment would have made analyses unnecessarily complex, and potentially biased. Many of the studies included in our meta-analysis sampled over multiple years, while others only covered shorter periods. As a consequence, averaged and non-averaged data would have been the result. In addition, and related to previous paragraph, adjusting also for sampling year would have required the calculation of odds ratio per country per year, making the results even more complicated and less parsimonious. In sum, we believe that while adding this level detail to the analysis may contribute to the overall quality of results, it would unfortunately render them fragmented.

3) Please consider using study year as a moderator of HIV prevalence throughout. I appreciate the example shown with PrEP in U.S.-based studies; but the rollout of ART, PEPFAR, and subsequent treatment-as-prevention policies likely have a greater effect on HIV prevalence than PrEP does, including among trans people. It would be helpful to sampling year(s) as a moderating variable throughout these analyses, for instance using 5-year periods (2000-2004; 2005-2009, etc) to categorize moderating effects. Without this, you cannot show trends in HIV infection in this population as clearly; and more importantly, it is hard to tell where we are now (compared to where we were 20 years ago).

We thank the reviewer for this suggestion. At the outset of this meta-analysis, we considered including study year as a moderator but decided not to do so for several reasons. Firstly, not every manuscript provides study year information and obtaining additional information on published studies from authors has proven to be difficult. Secondly, had we had been able to acquire that information (and our experience was that we could not), we would have ended up with many studies that do not fit in 5-year periods unanimously for all countries. The periods would then differ by country and also reduce, due to exclusion, sample sizes even more. In that sense, such an endeavor would make Reviewer 2’s comment (see below) even more valid. Thus, we aimed at maintaining the most robust data (in terms of sample size) with a global perspective. We agree that this limits the potential of the data to show trends in HIV infection as clearly as we would like to, but we had to work with the information that is available. If we had taken the approach suggested by the reviewer, we would have only been able to report data from the US, which is clearly not desirable from a global perspective. Nonetheless, we have now included not using study period as a moderator as a limitation on page 28 of the manuscript. 

4) Typo "indivi" on p. 25, line 121.

Thanks for showing us this typo. It has been corrected in the revised manuscript.

Reviewer 2

I reviewed the statistical approach used in this paper only. The approach that was used is suitable, with one caveat: In Table 3, sample sizes from countries outside the US appear too small to estimate prevalence. Can the authors comment on the acceptability of using such small samples?

We thank the reviewer for this comment. In the revised manuscript, we have added/expanded on small sample sizes as a limitation (see pg. 28). We do believe that including the data from countries outside of the US is necessary if we aim to provide a comprehensive and global picture of HIV prevalence and burden. Otherwise, the meta-analysis would be US-centric, and this would lead to different type of criticism. In this double trade-off situation, we believe that it is more important to report global data, and this is in line with previous approaches (see Baral et al., 2013). We hope the reviewer concurs.

---

## [Decision Letter · Decision Letter 1]

10 Jul 2020

PONE-D-20-01449R1

The worldwide burden of HIV in transgender individuals: An updated systematic review and meta-analysis

PLOS ONE

Dear Dr. Stutterheim,

Thank you for submitting your manuscript to PLOS ONE. After careful consideration, we feel that it has merit but does not fully meet PLOS ONE’s publication criteria as it currently stands. Therefore, we invite you to submit a revised version of the manuscript that addresses the points raised during the review process.

Please address major weaknesses identified by reviewer #1, in particular his comments # 1 & 2.

We look forward to receiving your revised manuscript.

Kind regards,

Chongyi Wei, DrPH

Academic Editor

PLOS ONE

Reviewers' comments:

Reviewer's Responses to Questions

**Comments to the Author**

1. If the authors have adequately addressed your comments raised in a previous round of review and you feel that this manuscript is now acceptable for publication, you may indicate that here to bypass the “Comments to the Author” section, enter your conflict of interest statement in the “Confidential to Editor” section, and submit your "Accept" recommendation.

Reviewer #1: (No Response)

2. Is the manuscript technically sound, and do the data support the conclusions?

Reviewer #1: Yes

3. Has the statistical analysis been performed appropriately and rigorously? 

Reviewer #1: Yes

4. Have the authors made all data underlying the findings in their manuscript fully available?

Reviewer #1: Yes

5. Is the manuscript presented in an intelligible fashion and written in standard English?

Reviewer #1: Yes

6. Review Comments to the Author

Reviewer #1: While the authors have spent substantial time and energy justifying their decisions, they have not chosen to incorporate the majority of the substantive recommendations provided. I do not find that their justifications are strong enough, especially regarding comparison populations and difficulty in moderating by study year, scientifically compelling. In fact the response to reviewers serves to distinguish and highlight an additional weakness that the comparison population (HIV prevalence in country-wide general populations) are all taken from 2017, which is not conceptually sound given that the samples included go back in some cases to 2000.

1) Please reconsider extracting sampling year(s) from each study, as best as is possible given that it may not be provided in some cases (please show from what studies this is not provided). That data can be incorporated into Table 1.

2) If not accepting a city-level comparison, please at least consider using study-specific, country-level data for the general population comparison group for the year (or, mean rate for multiple years if the sampling spans multiple years) that are best aligned with the study's sampling year(s), given the available UNAIDS and other country-level data.

3) Please reconsider moderation by study year/period. At the very least, consider using a dichotomous variable classifying moderation by pre- and post- widespread treatment-as-prevention adoption (e.g. ~2012 but will vary by country).

I think this manuscript has substantial potential for impact, which will really be heightened if the comparisons made, and the conclusions and implications that result from these, are less spurious.

7. PLOS authors have the option to publish the peer review history of their article (what does this mean?). If published, this will include your full peer review and any attached files.

Reviewer #1: No

---

## [Author Response · Author response to Decision Letter 1]

1 Dec 2020

PLEASE ADDRESS MAJOR WEAKNESSES IDENTIFIED BY REVIEWER #1, IN PARTICULAR HIS COMMENTS # 1 & 2.

1) PLEASE RECONSIDER EXTRACTING SAMPLING YEAR(S) FROM EACH STUDY, AS BEST AS IS POSSIBLE GIVEN THAT IT MAY NOT BE PROVIDED IN SOME CASES (PLEASE SHOW FROM WHAT STUDIES THIS IS NOT PROVIDED). THAT DATA CAN BE INCORPORATED INTO TABLE 1.

Thank you for this comment. Sampling years are included in Table 1. In previous versions, this information was included in Table 1 as well, in the final column with the title ‘Year of data collection’. In this revised version, we have moved this column to the left so that it is more evident. It is now placed next to the ‘Year of publication’ column.

2) IF NOT ACCEPTING A CITY-LEVEL COMPARISON, PLEASE AT LEAST CONSIDER USING STUDY-SPECIFIC, COUNTRY-LEVEL DATA FOR THE GENERAL POPULATION COMPARISON GROUP FOR THE YEAR (OR, MEAN RATE FOR MULTIPLE YEARS IF THE SAMPLING SPANS MULTIPLE YEARS) THAT ARE BEST ALIGNED WITH THE STUDY'S SAMPLING YEAR(S), GIVEN THE AVAILABLE UNAIDS AND OTHER COUNTRY-LEVEL DATA.

Thank you for this comment. As per the reviewer’s request, we have, in our revised manuscript, aligned the years of data collection for each study with country-level prevalence rates for the year(s) in which the data were collected. We have updated the methods section (lines 201-206) accordingly, outlining exactly how this was done. We indicate that we grouped studies by country and year of data collection. To calculate the HIV prevalence in the general population, we used the UNAIDS reports (for HIV frequency estimates) and US Census data (for population estimates) of the corresponding years. In addition, we indicate that we standardized prevalence rates instead of pooled prevalence rates, as this takes the weights for each study into account rather than crudely pooling prevalence rates across studies. Overall, this approach has less bias. 

We appreciate the reviewer re-emphasizing these analyses as the more fine-grained analyses do better justice to the data, and we now also mention this in our introduction (lines 110-112). The new analyses have yielded a higher odds ratio for both trans feminine individuals (66.0 versus 38.1 in our previous analysis) AND trans masculine individuals (6.6 versus 3.5). We have therefore adjusted the discussion section to reflect these higher odds ratios and their implications.

Also, as our initial intent was to, using analogous methodology, offer an updated meta-analysis that could be compared with previous meta-analyses, specifically Baral et al. (2013), we have, in our revision, opted not to discard the previous analysis but rather move it to the appendix so that readers can still compare those findings to previous meta-analytical findings. We trust the editor and reviewer agree with this transparency. We also believe that presenting both analyses contributes to both backward comparability and developing a novel standard of meta analytical reporting in this domain.

3) PLEASE RECONSIDER MODERATION BY STUDY YEAR/PERIOD. AT THE VERY LEAST, CONSIDER USING A DICHOTOMOUS VARIABLE CLASSIFYING MODERATION BY PRE- AND POST- WIDESPREAD TREATMENT-AS-PREVENTION ADOPTION (E.G. ~2012 BUT WILL VARY BY COUNTRY).

Thank you for this comment. We agree that such a test of moderation would be desirable, but we chose not to include it in the paper for several reasons. First of all, this analysis is much more complex than the data we are summarizing here: We cannot use country cascade data for the TasP moderation, but we would need cascade data for transgender individuals; such data is not widely available. Secondly, such a sufficiently large amount of data suitable for a meta-analysis would first be available only from the USA, and we value a global focus over a country specific: for some countries we do have only a limited amount of data and a dichotomization would lead to comparing single studies of different size with each other. Third, we ran a moderation analysis for PrEP (which is part of the bio-medical HIV prevention and treatment palette) and it did not yield strong positive findings. In fact, the trend is not yet reversed, but the amount of studies is also rather low. Robustly interpretable data for TasP and PrEP may not be available yet. Finally, the editor had stressed to focus on (1) and (2) of the comments, and we were not convinced, based on our findings, that this analysis could contribute much more to the larger picture. We have added a recommendation in the manuscript that reflects this comment and suggests its incorporation in future studies (lines 409-411). We trust this is acceptable.

---

## [Decision Letter · Decision Letter 2]

19 Aug 2021

PONE-D-20-01449R2

The worldwide burden of HIV in transgender individuals: An updated systematic review and meta-analysis

PLOS ONE

Dear Dr. Stutterheim,

Thank you for submitting your manuscript to PLOS ONE. After careful consideration, we feel that it has merit but does not fully meet PLOS ONE’s publication criteria as it currently stands. Therefore, we invite you to submit a revised version of the manuscript that addresses the points raised during the review process.

**
*I am really sorry for the delay that you have experienced in this paper. When I was invited to be the Editor this summer, it was really hard to secure reviewers. Please address the comments that both reviewers made.*
**

We look forward to receiving your revised manuscript.

Kind regards,

Viviane D. Lima

Academic Editor

PLOS ONE

Reviewers' comments:

Reviewer's Responses to Questions

**Comments to the Author**

1. If the authors have adequately addressed your comments raised in a previous round of review and you feel that this manuscript is now acceptable for publication, you may indicate that here to bypass the “Comments to the Author” section, enter your conflict of interest statement in the “Confidential to Editor” section, and submit your "Accept" recommendation.

Reviewer #1: All comments have been addressed

Reviewer #3: (No Response)

2. Is the manuscript technically sound, and do the data support the conclusions?

Reviewer #1: Yes

Reviewer #3: Yes

3. Has the statistical analysis been performed appropriately and rigorously? 

Reviewer #1: Yes

Reviewer #3: Yes

4. Have the authors made all data underlying the findings in their manuscript fully available?

Reviewer #1: Yes

Reviewer #3: Yes

5. Is the manuscript presented in an intelligible fashion and written in standard English?

Reviewer #1: Yes

Reviewer #3: Yes

6. Review Comments to the Author

Reviewer #1: I commend the authors on their responsiveness to the reviewer comments, which appear to have substantially strengthened this manuscript. Some minor notes for consideration:

1) Funding section should be removed from Methods and placed below Acknowledgments.

2) Because the studies analyzed span so many years, they do not represent current HIV prevalence in trans populations. So I recommend the authors note that as such, and use appropriate language in the abstract and the discussion (e.g., "over the course of the HIV epidemic, mean HIV prevalence among trans women has been X%" rather than "HIV prevalence in trans women is X%".

3) Some acknowledgment in limitations that moderation by year (including multi-year spans before and after countrywide TasP policies have been promulgated within specific countries) has not been performed, but is likely important for future research: HIV prevalence trends over time is an especially valuable analysis that will tell us a lot about which populations are falling through the structural/policy cracks.

I look forward to seeing this manuscript in print. Thank you for the opportunity to review.

Reviewer #3: This manuscript is a systematic review and meta-analysis of laboratory confirmed HIV prevalence for transgender populations globally, aimed to update the previously published review by Baral et. al, replicating the methodology. The study is well-conducted and represents an important contribution to the literature. However, there are a number of outstanding major and minor issues with the manuscript currently, as outlined below.

Major comments:

• “Sampling frame” is used incorrectly in the paper- the authors seem to be using this to refer to sampling method, however this term refers to the sampling framework or list of individuals in the population from which people were sampled into a given study.

• What is the rationale for including studies in this review published between 2000-2011- which overlaps directly with previously reviewed literature in the Baral et al (as well as Poteat et al) study the authors aimed to replicate? This should either be reconsidered, or the value of the approach explicitly addressed in the manuscript.

• While inclusion of both the pooled prevalence and standardized prevalence estimates by the authors is a laudable contribution to the field of transgender health research and HIV, the authors do not discuss the large discrepancies in the resulting estimates by method. A discussion of the results by method is warranted, particularly given the wide range in estimates produced by the two methods. Authors should provide context for why they believe the estimate vary to such a degree, and issues related to validity and reliability.

• A number of studies reporting laboratory confirmed HIV prevalence data for transgender populations that were captured in the previous reviews in this area detailed in the current manuscript seem to be missing from the current review. These include Rich et al. 2017 Culture, Health and Sexuality with data on transgender men, and several studies of transgender women.

• The authors reference critiques of previously published reviews in this area, including the critique that the Baral et al. estimates may have been biased by inclusion of multiply marginalized samples of transgender women with heightened HIV risk factors in pooled prevalence estimation, citing this as a motivation for the sub-analysis in the current study by sampling method. However, this critique would be best addressed by sub-analysis by sample sub-population. An additional sub-analysis of studies captured in this review by sample sub-population (e.g. transgender sex workers, etc.) would be a major contribution to the literature. If not, a more robust discussion of the absence of such an analysis should be included in the limitations section.

7. PLOS authors have the option to publish the peer review history of their article (what does this mean?). If published, this will include your full peer review and any attached files.

Reviewer #1: No

Reviewer #3: No

---

## [Author Response · Author response to Decision Letter 2]

28 Sep 2021

REVIEWER 1

I commend the authors on their responsiveness to the reviewer comments, which appear to have substantially strengthened this manuscript. Some minor notes for consideration:

Thank you for your encouragement.

1) Funding section should be removed from Methods and placed below Acknowledgments.

Thank you. We have now moved the funding section to below the acknowledgements (lines 492-495).

2) Because the studies analyzed span so many years, they do not represent current HIV prevalence in trans populations. So I recommend the authors note that as such, and use appropriate language in the abstract and the discussion (e.g., "over the course of the HIV epidemic, mean HIV prevalence among trans women has been X%" rather than "HIV prevalence in trans women is X%".

Thank you for drawing our attention to the need to hedge and frame claims about prevalence rates. As requested, we have revised the abstract and discussion to reflect this more nuanced claim. We have also reviewed the remainder of the manuscript to ensure that this is abundantly clear throughout. We trust this is now in order. 

3) Some acknowledgment in limitations that moderation by year (including multi-year spans before and after countrywide TasP policies have been promulgated within specific countries) has not been performed, but is likely important for future research: HIV prevalence trends over time is an especially valuable analysis that will tell us a lot about which populations are falling through the structural/policy cracks.

Thank you for this comment. As indicated in previous rounds of revisions, we agree that a test of moderation would be desirable, but we chose not to include it in the paper for several reasons. First, this analysis is much more complex than the data we are summarizing here: It is not possible to use country cascade data for a TasP moderation; rather, we would need cascade data for transgender individuals, and that data is not widely available. Second, such a sufficiently large amount of data suitable for a meta-analysis would, at this point in time, only be available for the US, and we value a global focus over a country specific focus. For many countries, we have only a limited amount of data and a dichotomization would lead to comparing single studies of different sizes with one another other. Third, we ran a moderation analysis for PrEP (which is part of the bio-medical HIV prevention and treatment palette) and it did not yield strong positive findings. In fact, the trend is not yet reversed, but the amount of studies is also rather low. We thus believe that robustly interpretable data for TasP and PrEP is not yet available. We have added a recommendation in the manuscript that reflects this (lines 437-442) and we trust that this adequately addresses this comment.

I look forward to seeing this manuscript in print. Thank you for the opportunity to review.

Thank you! We too are very much looking forward to having this paper in print.

REVIEWER 3

This manuscript is a systematic review and meta-analysis of laboratory confirmed HIV prevalence for transgender populations globally, aimed to update the previously published review by Baral et. al, replicating the methodology. The study is well-conducted and represents an important contribution to the literature. However, there are a number of outstanding major and minor issues with the manuscript currently, as outlined below.

Thank you for your positive comments and feedback. We agree that this is an important contribution to the literature and hope to have adequately addressed your concerns in this revision.

• “Sampling frame” is used incorrectly in the paper- the authors seem to be using this to refer to sampling method, however this term refers to the sampling framework or list of individuals in the population from which people were sampled into a given study.

Thank you for this comment. We have removed the term sampling frame and replaced it with sampling method throughout the paper.

• What is the rationale for including studies in this review published between 2000-2011- which overlaps directly with previously reviewed literature in the Baral et al (as well as Poteat et al) study the authors aimed to replicate? This should either be reconsidered, or the value of the approach explicitly addressed in the manuscript.

Thank you for this comment. As now explicated in the revised manuscript (lines 132-142), the rationale for our decision to include data presented in previous meta-analyses was: 1) Meta analyses have the potential to become more robust if more data is included. Instead of providing a disconnected pattern of meta-analytic summaries across a number of meta-analyses (with the danger of limited overlap), we chose to cover the whole period of studies available in order to provide a complete, comprehensive, and also nuanced understanding of the worldwide prevalence and burden of HIV among transgender individuals (line 134-136); 2) Because we presented two different approaches in our meta-analysis, it was imperative that we include the data that has been covered by previous meta-analyses. If those data were not included, there would be no evidence indicating if the differences found also hold based on data included in previous meta-analytic summaries. 

• While inclusion of both the pooled prevalence and standardized prevalence estimates by the authors is a laudable contribution to the field of transgender health research and HIV, the authors do not discuss the large discrepancies in the resulting estimates by method. A discussion of the results by method is warranted, particularly given the wide range in estimates produced by the two methods. Authors should provide context for why they believe the estimate vary to such a degree, and issues related to validity and reliability.

Thank you for this comment. It is important to note that the standardized prevalence estimate was only applied for the overall prevalence estimation and we considered this to be methodologically more sound than a pooled prevalence estimate. In a pooled estimate, the total study population and total HIV cases are summed and then a crude proportion is calculated. This does not take heterogeneity and variation among the included studies into account. Our standardization approach entailed taking the weights from each country-year into account. Without the weighted standardization, a country-year combination that contains large study samples such as US-2010 (see Table 2; n=16,943 trans feminine individuals), or a country-year that includes small samples such as the Netherlands-2003 (Table 2; n=61 trans feminine individuals), may deliver misleading pooled results. The standardized prevalence by the weight from each country-year thus delivers a more robust estimation because it accounts for variations in study settings, sample sizes, and data quality. This argumentation is now included in the discussion section (lines 360-368).

• A number of studies reporting laboratory confirmed HIV prevalence data for transgender populations that were captured in the previous reviews in this area detailed in the current manuscript seem to be missing from the current review. These include Rich et al. 2017 Culture, Health and Sexuality with data on transgender men, and several studies of transgender women.

Thank you for your attention to detail here. We have gone back over previous reviews to ascertain if studies were missing. We established that five studies listed in those reviews were not included in our meta-analyses: 1) Bokhari et al., 2007; 2) Spizzichino et al., 2001; 3) Shrestha et al., 2011; 4) Simon et al., 2000; and 5) Rich et al., 2017. The first four were not included because they reported duplicate data. Their samplies overlapped with, respectively: 1) Khan et al., 2008; 2) Zacarelli et al., 2004; 3) Schulden et al., 2008; and 4) Reback et al., 2005. Their exclusion is in line with the procedure outlined in our methods section. To wit: "When studies reported duplicate data, the study with the smallest sample size was excluded." With respect to Rich et al., 2017, this was an oversight. Apologies. The study had a qualitative design and reported a prevalence rate of 0 with a sample of merely 11 participants. During data extraction, this was inadvertently allocated as an article not to be included. We have now corrected this. We have now included Rich et al., 2017 and rerun the analyses for trans masculine individuals. The manuscript has been adjusted to reflect this analyses and this can be seen in the abstract (lines 30-36) and the results section (lines 248-255, Table 3, and Figure 3). Overall, the conclusions are not impacted by the inclusion of the 11 additional participants in Rich et al., 2017. 

• The authors reference critiques of previously published reviews in this area, including the critique that the Baral et al. estimates may have been biased by inclusion of multiply marginalized samples of transgender women with heightened HIV risk factors in pooled prevalence estimation, citing this as a motivation for the sub-analysis in the current study by sampling method. However, this critique would be best addressed by sub-analysis by sample sub-population. An additional sub-analysis of studies captured in this review by sample sub-population (e.g. transgender sex workers, etc.) would be a major contribution to the literature. If not, a more robust discussion of the absence of such an analysis should be included in the limitations section.

We appreciate the reviewer’s desire to optimize this manuscript. However, we believe that it is important that we collectively acknowledge the various demands that have been placed on this manuscript during the editorial process. Throughout this process, we have differentiated the analyses along a number of lines. Initially, this was along the lines of trans feminine and trans masculine populations, as well as geographical regions, sampling methods, and the introduction of PrEP. Later, we were asked to look at sampling year for each study, which we did. We were also asked to look by city, which was not feasible (see previous responses to reviewers and the limitations section). In this round, we have additionally been asked to differentiate analyses along the lines of sub-populations within the trans community (i.e., sex workers). Throughout the editorial process, we have accommodated the requests for which sufficient data is available on a global scale. While we acknowledge that it would be interesting to look at subgroups, the primary level data does not yet allow for all such analyses on a global scale. The subpopulation analysis now being requested would, similarly to the moderation by TasP analysis requested, be focused exclusively on US data which is not the global perspective we set out to take in this manuscript. With this in mind, we expanded the discussion section to include the following: “our analysis… did not separately ascertain prevalence rates for trans feminine individuals who engage in sex work versus those who do not as primary level data on this is not available on a global scale. We recommend that future research take these potential shortcomings into consideration. Specifically, we recommend that future research explicitly investigate prevalence among sub-populations within the transgender community… as this will provide an even more comprehensive picture of HIV prevalence and burden among transgender individuals.” (lines 435-442).

---

## [Editor Report · Decision Letter 3]

3 Nov 2021

The worldwide burden of HIV in transgender individuals: An updated systematic review and meta-analysis

PONE-D-20-01449R3

Dear Dr. Stutterheim,

We’re pleased to inform you that your manuscript has been judged scientifically suitable for publication and will be formally accepted for publication once it meets all outstanding technical requirements.

Kind regards,

Viviane D. Lima

Academic Editor

PLOS ONE
---

## [Editor Report · Acceptance letter]

10 Nov 2021

PONE-D-20-01449R3 

The worldwide burden of HIV in transgender individuals: An updated systematic review and meta-analysis 

Dear Dr. Stutterheim:

I'm pleased to inform you that your manuscript has been deemed suitable for publication in PLOS ONE. Congratulations! Your manuscript is now with our production department. 

Kind regards, 

on behalf of

Dr. Viviane D. Lima 

Academic Editor

PLOS ONE